



# An upper mesopelagic zone carbon budget for the subarctic North Pacific

Brandon M. Stephens[1,2*], Montserrat Roca-Martí[3*], Amy E. Maas[4], Vinícius J. Amaral[5], Samantha Clevenger[6], Shawnee Traylor[6,7], Claudia R. Benitez-Nelson[8], Philip W. Boyd[9], Ken O. Buesseler[7], Craig A. Carlson[1], Nicolas Cassar[10,11], Margaret Estapa[12], Andrea J. Fassbender[13], Yibin Huang[14], Phoebe J. Lam[5], Olivier Marchal[15], Susanne Menden-Deuer[16], Nicola L. Paul[1], Alyson E. Santoro[1], David A. Siegel[17], David P. Nicholson[7]

[1]Department of Ecology, Evolution and Marine Biology, University of California, Santa Barbara, CA 93106, USA
[2]Institute of Oceanography, National Taiwan University, Taipei, Taiwan
[3]Institut de Ciència i Tecnologia Ambientals (ICTA-UAB), Universitat Autònoma de Barcelona, Cerdanyola del Vallès, Spain
[4]Bermuda Institute of Ocean Sciences, School of Ocean Futures, Arizona State University, St. Georges, GE01, Bermuda
[5]Department of Ocean Sciences, University of California Santa Cruz, US
[6]The MIT-WHOI Joint Program in Oceanography/Applied Ocean Science and Engineering, Cambridge and Woods Hole, MA, USA
[7]Marine Chemistry and Geochemistry Department, Woods Hole Oceanographic Institution, Woods Hole, MA 02543, USA
[8]School of the Earth, Ocean & Environment, University of South Carolina, Columbia, SC 29208, USA
[9]Institute for Marine and Antarctic Studies, University of Tasmania, Hobart, Tasmania 7001, Australia
[10]Division of Earth and Climate Sciences, Nicholas School of the Environment, Duke University, Durham, NC 27708, USA
[11]CNRS, Université de Brest, IRD, Ifremer, LEMAR, F-29280 Plouzané, France
[12]School of Marine Sciences, Darling Marine Center, University of Maine, Walpole, ME 04573, USA
[13]NOAA/OAR Pacific Marine Environmental Laboratory, Seattle, WA, USA
[14]State Key Laboratory of Marine Environmental Science, College of Ocean and Earth Sciences, Xiamen University, Xiamen, FJ 350800, China
[15]Department of Geology and Geophysics, Woods Hole Oceanographic Institution, Woods Hole, MA 02543, USA
[16]Graduate School of Oceanography, University of Rhode Island, RI 02882, USA
[17]Earth Research Institute and Department of Geography, University of California, Santa Barbara, CA 93106, USA

*These authors contributed equally to this article.

*Correspondence to*: Brandon M. Stephens (bstephens@ntu.edu.tw), Montserrat Roca-Martí (montserrat.roca.marti@uab.cat), David P. Nicholson (dnicholson@whoi.edu)

**Abstract.** Mesopelagic zone (MZ) carbon budgets comparing supply with demand can be difficult to constrain due to the temporal and spatial offsets between key sources and sinks, and due to nuances of the measurement techniques used, their associated uncertainties, and potential sampling biases. To address some of these challenges, the EXport Processes in the Ocean from RemoTe Sensing (EXPORTS) campaign increased the number and variety of simultaneous measurements to monitor temporal variability in the MZ carbon budget using both a Lagrangian frame of reference and long-term autonomous collection. In this study, we collate a comprehensive combination of new and previously published organic carbon supply and demand measurements collected from the surface (5 m) to the upper MZ, defined here as depths from 100 m to 500 m. Research cruise-based measurements were collected near the subarctic North Pacific's Ocean Station Papa (OSP) during the August 2018 EXPORTS field campaign. The supply of particulate organic carbon (POC) into the MZ averaged 3.0 mmol C m$^{-2}$ d$^{-1}$,



with roughly equal contributions from passively sinking particles and deposits from active diel vertical migration of zooplankton. MZ carbon demand, in the form of respiration, averaged 5.7 mmol C m$^{-2}$ d$^{-1}$, with most of this demand from free-living bacterioplankton and minor contributions from zooplankton and particle-attached bacterioplankton. The ship-based estimate of water column demand exceeded ship-based supply. Moreover, the MZ carbon demand may have been even higher based on trends in dissolved oxygen concentration from a glider and a biogeochemical float operating from August to November 2018. This imbalance could be resolved by particle dynamics influencing timescales of organic carbon utilization prior to the field campaign. Net community production (NCP) rates measured during the preceding spring and early summer of 2018 based on long-term mooring estimates of dissolved inorganic carbon concentrations. Seasonal trends in upper MZ backscattering measurements in the vicinity of OSP, in addition to long-term decreases in dissolved organic carbon, suggest that the excess in organic C demand in the upper MZ could be accounted for by the release, disaggregation, and subsequent slow degradation of particles from NCP earlier in the year. The OSP MZ carbon budget presented here demonstrates that studies attempting to constrain the fate of exported POC require the integration of samples over short-time (days to weeks; ships) and long-time (months; remote observations) scales. Finally, based on this carbon mass balance approach, we highlight that studies attempting to validate carbon dioxide removal through particle export should consider comparing multiple sample collections and monitoring over longer time scales.

**Short Summary.** The ocean's mesopelagic zone (MZ) plays a crucial role in the global carbon cycle. This study combines new and previously published measurements of organic carbon supply and demand collected in August 2018 for the MZ in the subarctic North Pacific Ocean. Supply was insufficient to meet demand in August, but supply entering into the MZ in the spring of 2018 could have met the August demand. Results suggest observations over seasonal time scales may help to close MZ carbon budgets.

# 1 Introduction

Quantifying the fate and storage of recently-fixed $CO_2$ below the euphotic zone is vital for budgets of the global carbon cycle and constraining the role of oceans as a carbon sink (e.g., Boyd, 2015; Huang et al., 2023). The strength of the biological carbon pump, as measured by the downward flux of organic C out of the euphotic zone, further influences the quantity of oxygen utilized in the ocean's interior (Suess, 1980), which can impact deep ocean communities and the size of oxygen minimum zones (Breitburg et al., 2018). As marine carbon dioxide removal (CDR) strategies are being envisioned, there is a continued need to constrain uncertainties in tracking the fate of exported carbon and to identify the impacts of such strategies on the oceans (e.g., Boyd et al., 2022).

Sources of organic carbon to the mesopelagic zone (MZ) include both passively sinking particles and active flux (e.g., migrating zooplankton and/or fish), but can also include non-sinking organic matter delivered by particle disaggregation,



carbon delivered via vertical mixing (Omand et al., 2015; Boyd et al., 2019; Siegel et al., 2023), and dark carbon fixation (Reinthaler et al., 2010). Carbon demand (including respiration) in the dark ocean mainly comes from bacterioplankton (bacteria+archaea) and micro- and mesozooplankton (Collins et al., 2015; Iversen, 2023). Measuring supply and demand rates in the MZ can be challenging due to methodological constraints, carbon conversion uncertainties, and the timescales over

75 which various measurements integrate (Burd et al., 2010; Iversen, 2023; Herndl et al., 2023). Combining multiple methodological approaches, each with its inherent though independent assumptions, integration timescales, and errors, should, in principle, lead to a better constraint on the fate of organic carbon exported out of the euphotic zone (EZ).

Early efforts demonstrated that the passive sinking flux of particles caught by sediment traps alone was insufficient to meet

MZ communities' carbon utilization requirements. For instance, a MZ carbon budget for the North Pacific's Ocean Station Papa (OSP) found that sinking particle flux attenuation was much lower than predicted bacterioplankton carbon demand (BCD, Boyd et al., 1999), which is defined as biomass production plus respiration. Studies at the Bermuda Atlantic Time-series Study site (BATS) and Station ALOHA found that dissolved organic carbon (DOC) released by vertically migrating zooplankton can provide a significant and variable source of organic carbon, potentially exceeding supply from sinking particles (Steinberg

et al., 2000; Steinberg et al., 2008; Xiang et al., 2023). However, the measured sources were still insufficient to meet the predicted BCD. More recently, a balanced carbon budget, within margins of error, has been derived for the Porcupine Abyssal Plain site in the North Atlantic, suggesting that sinking carbon routed through zooplankton into bacteria via sloppy feeding and other mechanisms (Giering et al., 2014; Baumas et al., 2023), was sufficient to meet the measured BCD.

Understanding and quantifying organic carbon export is of central importance to the National Aeronautics and Space Administration (NASA)-supported EXport Processes in the Ocean from RemoTe Sensing (EXPORTS) Program. The overall study was designed to contrast particle size and export efficiency (i.e., the ratio of export production to NPP at the base of the euphotic zone) across two distinct regions: stratified summer conditions in the eastern Subarctic North Pacific and less-stratified, post-bloom spring conditions in the North Atlantic (Siegel et al., 2016). The following study presents a carbon

budget for the upper MZ from the North Pacific EXPORTS field campaign, based on an unprecedented suite of simultaneous upper ocean (0-500 m) carbon supply and demand measurements covering a range of timescales.

Previously published results from the EXPORTS North Pacific field campaign, based at OSP, captured relatively warmer stratified waters and minima in surface macronutrient concentrations (Siegel et al., 2021), which are typical late summer

physical conditions for the study region (Whitney and Freeland, 1999; Harrison, 2002). Within the EZ, biological conditions were typical for OSP (Boyd and Harrison, 1999), with relatively low concentrations of surface chlorophyll-a (Chl-a) and particulate organic carbon (POC), as well as low net primary production (NPP) dominated by small phytoplankton cells (Meyer et al., 2022, Graff et al., 2023). Net community production (NCP), defined as the difference between gross primary production and respiration, was slightly positive (Niebergall et al., 2023), and phytoplankton growth was well balanced with





microzooplankton grazing, suggesting microzooplankton grazing to be the dominant loss process (McNair et al., 2021). Rates of transfer organic carbon between production and assimilation were relatively balanced, leaving little organic carbon available for export (McNair et al., 2023). Bacterioplankton had relatively high growth efficiencies (31% reported in Stephens et al., 2020), similar to prior summer estimates for OSP (Sherry et al., 1999); its growth was likely supported by phytoplankton-derived dissolved organic matter (DOM) "sourced" from gross carbon production and zooplankton sloppy feeding (Stephens et al., 2023).

Below the EZ, POC sinking fluxes were also relatively low, with an export efficiency of 10-13%, similar to previous late summer estimates at the study site (Buesseler et al., 2020; Estapa et al., 2021). Maximum POC flux was 5.5 mmol C m$^{-2}$ d$^{-1}$ at 50 m, decreasing rapidly with depth within the EZ and below (Buesseler et al., 2020). Although diatom contributions to POC export were also relatively low (9-13%), a high fraction (about 33%) of biogenic silica was exported out of the upper 100 m compared with other high-nutrient, low-chlorophyll regions (Brzezinski et al., 2022). Gel trap collectors demonstrated that sinking particles shifted from relatively high contributions of salp fecal pellets to a variable contribution of salp, long, and small fecal pellets (Durkin et al., 2021), the attenuation of which may have been influenced in part by particle-attached bacterioplankton community succession patterns (Stephens et al., 2024).

Between the mixed layer depth (MLD) and 500 m, suspended and sinking POC collected using marine snow catchers were found to be of similar size, but suspended POC contained three-fold greater transparent exopolymer particles as compared with sinking POC (Romanelli et al., 2023), likely contributing to the buoyancy of suspended POC. Further evidence based on compound-specific stable isotope analysis of amino acids suggested that small (< 6 μm) suspended or slowly sinking particles and, to a lesser extent, organic matter delivered by vertically migrating zooplankton were likely key organic C sources supporting mesopelagic communities (Shea et al., 2023; Wojtal et al., 2023). A seasonal time series of profiling float data at OSP, initiated immediately after the 2018 EXPORTS cruise, suggest that estimates of POC concentrations based on the light backscattering coefficient (bbp) increased just below 100 m in the spring of 2019, thereby potentially acting as an additional carbon supply source to the upper MZ (Huang et al., 2022).

Building on previously published results for OSP and EXPORTS, this study presents an organic carbon budget based on the supply and demand within the "upper MZ," which we define here as between 100 and 500 m. Given that the MZ extends deeper into the ocean (e.g., 1,000 m), we emphasize that observations collected between 100 and 500 m are specifically representative of the "upper" MZ. The primary focus is on measurements collected during the August 2018 occupation of OSP as part of EXPORTS, but these data are supplemented by glider, float, and profile observations collected before and after this period. To reduce errors due to double counting carbon utilization within the upper MZ, we have adopted the approach of Giering et al. (2014) by focusing solely on respiration, as opposed to respiration+production. Though focusing on respiration may underestimate the total carbon demand, double counting could occur if a particle enters the MZ, is consumed by an



organism, and then is consumed again by a detritivore. The following data analysis suggests that studies attempting to constrain
the fate of exported POC should collect field observations over both short- and long-term scales, with implications for
optimizing observation strategies for deep ocean CDR.

## 2 Methods and Results

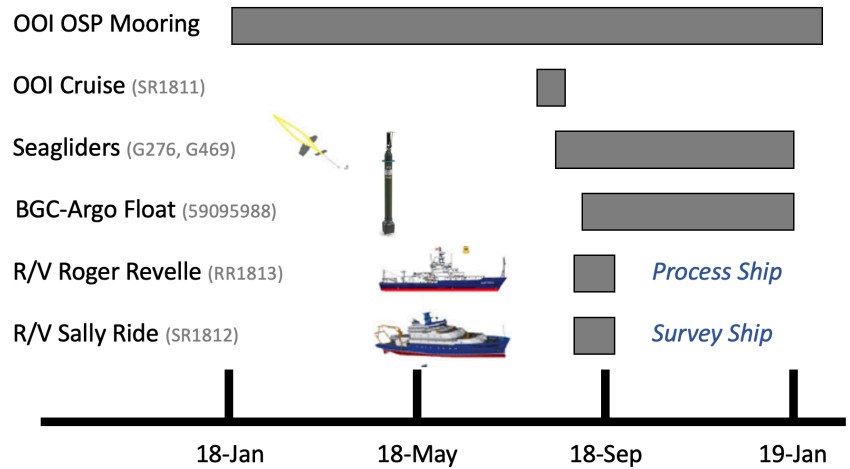

**Figure 1: Sampling periods before, during, and after the August 2018 EXPORTS field campaign using a variety of**
**sampling platforms. After Siegel et al. (2021).**

The EXPORTS North Pacific field program sampled a large mesoscale eddy between August 14th and September 9th, 2018,
using two ships and autonomous sampling platforms (Fig. 1) stationed near OSP (50°N, 145°W, Siegel et al., 2021). Shipboard
data presented here were collected from the R/V Sally Ride (SR1812, the "survey" ship) and R/V Roger Revelle (RR1813, the
"process" ship). The process ship followed a drifting Lagrangian float deployed at approximately 100 m while the survey ship
conducted spatial surveys around the process ship. Water casts typically included 12 standard depths at 5, 20, 35, 50, 65, 80,
95, 120, 145, 195, 330 and 500 m. Three profiles were collected from the Ocean Observatories Initiative (OOI)-supported
cruise (SR1811) on July 27th near OSP for DOC concentrations. The mixed layer and euphotic zone (0.1% of surface
photosynthetically active radiation) depths averaged $29 \pm 4$ m and $95 \pm 11$ m during the cruise period, respectively (Siegel et
al., 2021).

Continuous surface observational data were collected from the National Oceanic and Atmospheric Administration Pacific
Marine Environmental Laboratory's OOI OSP surface mooring from January through December 2018. Seagliders were
deployed on July 23rd (G276 and EXPORTS glider) and August 21st (G469). A biogeochemical (BGC) Argo profiling float
(WMO ID: 59095988) was deployed on August 16th, 2018. Both the BGC float and Seagliders collected data into January





2019. The BGC float, mooring, and gliders continuously collected temperature, salinity, oxygen and nitrate concentrations, particle backscattering coefficient, chlorophyll fluorescence, and pH, allowing for the estimation of dissolved inorganic carbon (DIC) concentrations.

Several of the datasets presented here have been previously published, and the uncertainties surrounding conversion factors are discussed in those publications (e.g., as highlighted above). Therefore, only a brief description of methods is included here. The present study's goal is to evaluate the combined rates of organic carbon supply and demand and elucidate the implications of carbon conversion factors and uncertainty estimates.

## 2.1 Carbon Supply to the Mesopelagic Zone

### 2.1.1 Passive Flux from Sinking Particles

During the August 2018 EXPORTS cruise, POC flux attenuation (i.e., removal) with depth within the upper MZ was used to estimate carbon supply from passively sinking particles using four separate methodologies during three-time intervals, or "Epochs." Epoch 1 spanned August 14th to August 23rd, Epoch 2 spanned August 23rd to August 31st, and Epoch 3 spanned August 31st to September 9th, 2018.


POC fluxes were directly measured using surface-tethered and neutrally-buoyant sediment traps deployed for 3-6 days at five depths in the upper MZ during each Epoch, as described in detail by Estapa et al. (2021). Cruise-mean POC fluxes measured with sediment traps decreased from $1.4 \pm 0.5$ mmol C m$^{-2}$ d$^{-1}$ (mean $\pm$ propagated standard error) at 95-105 m to $0.9 \pm 0.2$ mmol C m$^{-2}$ d$^{-1}$ at 500-510 m (Fig. 2a), resulting in a flux attenuation of $0.5 \pm 0.5$ mmol C m$^{-2}$ d$^{-1}$. Most flux attenuation
occurred between 95 and 155 m (66%), with negligible or minimal attenuation occurring below 200 m. Epoch-mean POC fluxes increased 2- to 3-fold at 500 m from Epoch 1 to Epochs 2 and 3, and fluxes increased 2- to 3-fold at 95-105 m and 145-155 m from Epochs 1 and 2 to Epoch 3. This variability in POC fluxes between Epochs resulted in depth-integrated flux attenuation estimates ranging from 0 mmol C m$^{-2}$ d$^{-1}$ (in Epoch 2) to 1.0 mmol C m$^{-2}$ d$^{-1}$ (in Epoch 3; Fig. 2a).

Thorium-234 ($^{234}$Th) was used to derive POC fluxes in the upper MZ based on seawater profiles collected at high spatial and temporal resolution in combination with the POC/$^{234}$Th ratio determined in size-fractionated particles (Buesseler et al., 2020; Roca-Martí et al., 2021). Cruise-mean POC fluxes derived from a steady-state and a non-steady-state $^{234}$Th-based models decreased from $2.0 \pm 0.6$ mmol C m$^{-2}$ d$^{-1}$ at 100 m to $1.2 \pm 1.2$ mmol C m$^{-2}$ d$^{-1}$ at 500 m (Fig. 2b), resulting in a flux attenuation of $0.8 \pm 1.3$ mmol C m$^{-2}$ d$^{-1}$. The most substantial flux attenuation in the MZ occurred between 100 and 150 m (84%). Similar
to traps, POC fluxes estimated by the $^{234}$Th method reached a minimum of 200 m and did not decrease significantly below that depth.



**Figure 2: Epoch and cruise-mean profiles for carbon supply (subplots a to h) and demand (subplots i to l) estimates. Subplots a to d and i to j represent discrete sample collections, whereas subplots e to h and k to l represent samples collected across depth intervals. Subplots e to h are presented as depth-integrated units for comparison with passive supply subplots a to d. Net primary production (NPP) from inversion (subplot h) represents supply estimated from the inversion of size-fractionated [POC] data (Amaral et al., 2022), and only those data for the MZ are shown. Subplots i to l are shown in volumetric units. Community respiration from the glider and biogeochemical (BGC) float (subplot i) was estimated from long-term (>90 days) changes in [O₂] and converted to carbon units using a 1.4:1 O₂:C conversion. Note differences in the x-axis scales, which were adjusted to show depth variation. See Appendix 1 for the datasets shown here.**



The polonium-210 ($^{210}$Po) method was used to derive POC fluxes in a similar manner to $^{234}$Th at three sampling stations (Roca-Martí et al., 2020). $^{210}$Po measurements resulted in cruise-mean POC fluxes that were similar to $^{234}$Th at 100 m (2.6 ± 0.3 mmol C m$^{-2}$ d$^{-1}$, Fig. 2c) but 3 times lower at 500 m (0.4 ± 0.2 mmol C m$^{-2}$ d$^{-1}$), leading to a higher flux attenuation of 2.2 ± 0.4 mmol

C m$^{-2}$ d$^{-1}$. Overall, the $^{210}$Po method showed (i) a decrease in the flux attenuation with increasing depth in the mesopelagic zone (as do the other passive sinking flux methods), but (ii) showed greater flux attenuation between 330 and 500 m (0.4 mmol C m$^{-2}$ d$^{-1}$).

POC fluxes were also estimated by imaging and classifying sinking particles collected in polyacrylamide gel trap collectors

(Durkin et al., 2021). POC fluxes across all sinking particle categories decreased from 2.2 ± 0.7 mmol C m$^{-2}$ d$^{-1}$ at 95-105 m to 0.7 ± 0.2 mmol C m$^{-2}$ d$^{-1}$ at 500-510 m (Fig. 2d), resulting in a flux attenuation of 1.5 ± 0.7 mmol C m$^{-2}$ d$^{-1}$, which was contributed entirely by attenuation within the 95-155 m layer. Similar to estimates from sediment traps as mentioned above (Estapa et al., 2021), the POC fluxes measured by gel traps showed the lowest and highest flux attenuation in Epochs 2 (0.4 mmol C m$^{-2}$ d$^{-1}$) and 3 (2.2 mmol C m$^{-2}$ d$^{-1}$), respectively (Fig. 2d).


The four methods described above result in an average transfer efficiency of sinking POC between 100 and 500 m of 42 ± 22%, which means that about 40% of the POC flux out of the euphotic zone was transferred throughout the upper MZ and escaped the lower boundary of 500 m.

**2.1.2 Active Flux from Migrating Zooplankton**

Typically, the active flux of carbon from the upper ocean to the MZ from migrating zooplankton is the sum of excretion/egestion (DOC and POC), mortality, and the respiration of organic carbon consumed in the surface ocean but respired at depth (e.g., Steinberg et al., 2000). The active flux was estimated here based on a combination of rates of DOC excretion, fecal pellet production, and predation-based mortality for vertically migrating zooplankton, as previously published from the

EXPORTS program at OSP (Stamieszkin et al., 2021; Maas et al., 2021a; Steinberg et al., 2023). Active flux by migrators appeared to increase slightly over the five depth bins from which samples were collected (Fig. 2e-g), and there was an acoustically observed biomass peak in migrators at the 0.001 μmol m$^{-2}$ s$^{-1}$ isolume near ~300 to 350 m (Omand et al. 2021). When vertically integrated (100 to 500 m), cruise-averaged rates of DOC excretion and fecal pellet production were 0.2 mmol C m$^{-2}$ d$^{-1}$ and 0.1 mmol C m$^{-2}$ d$^{-1}$, respectively (Fig. 2f-g). The vertically-integrated cruise-averaged rate of predation-based

mortality was 0.2 mmol C m$^{-2}$ d$^{-1}$ (Fig. 2e). These three fluxes resulted in a total combined active flux of 0.5 ± 0.1 mmol C m$^{-2}$ d$^{-1}$. The contribution of salps to total active flux was highest early in the cruise, contributing up to 29% of the active flux in Epoch 1, but was relatively small when averaged over the entire cruise period (12%).



### 2.1.3 Temporal DOC Changes

There was a systematic decrease in DOC concentrations between 95 and 500 m depths from July 27[th] to August 17[th] to September 7[th], 2018 (42 total days between profile collections). If we assume that the decrease in DOC results from removal via respiration, we can constrain the DOC removal rate at seven discrete depths. DOC removal rates decreased from 0.05 mmol C m$^{-3}$ d$^{-1}$ at 95 m to 0.01 mmol C m$^{-3}$ d$^{-1}$ at 500 m (Supp. Fig. 1). The highest DOC removal rates, 0.05 and 0.02 mmol C m$^{-3}$ d$^{-1}$, were found between 95 and 145 m. These depths coincide with the most POC flux attenuation, supporting the notion that

organic carbon was rapidly recycled at these depths. DOC concentration differences for samples collected at 95 m between July and September (53.7 and 52.0 mmol C m$^{-3}$, respectively) were more than twice the mean instrumental uncertainty (1.7 mmol C m$^{-3}$ versus mean coefficient of variation (CV) of 1.3%, resulting in 2x instrumental uncertainty of 1.4 mmol C m$^{-3}$ at 95 m). Below 110 m, DOC concentrations decreased, but differences at all depths were less than 2x mean instrumental precision of 1.3% CV. Additionally, t-tests comparing the DOC concentrations between July and September showed these

concentrations to be not significantly different (p > 0.05) for all depths. Thus, long-term DOC changes in the upper MZ are difficult to constrain due to high methodological uncertainties. Yet, decreasing concentrations at all depths suggest a net DOC contribution to upper MZ demand. When integrated over the 95 to 500 m depth range, DOC removal rates averaged 5.6 ± 3.2 mmol C m$^{-2}$ d$^{-1}$.

### 2.1.4 Chemoautotrophic production

Chemoautotrophic production can be estimated from nitrification rates measured during the cruise (Santoro et al., manuscript in prep) using a ratio of DIC uptake to nitrification of 1:10 (Bayer et al., 2023; Reinthaler et al., 2010). Upper MZ-integrated (95-500 m) chemoautotrophy rates were estimated to be 0.1 ± 0.1 mmol C m$^{-2}$ d$^{-1}$. Other processes, like carbon fixation on non-sinking particles, could supply carbon to the upper MZ (e.g., Baltar et al., 2010). However, contributions from this process

are less constrained and were not estimated during the cruise.

### 2.1.5 Net Community Production

During the EXPORTS cruise period, NCP from 0 to 100 m depth was calculated using a mass balance approach from a merged dissolved oxygen concentration record from three gliders (two Ocean Observatories Initiative Slocum Gliders and one

Seaglider) for the cruise period from August 14[th] to September 7[th] (Traylor et al., submitted). Based on this calculation, NCP, as integrated over the upper 100 m, averaged 3.8 ± 0.6 mmol C m$^{-2}$ d$^{-1}$, which aligns with findings by Niebergall et al. (2023) collected during the OSP-based EXPORTS cruise.





To estimate long-term (December 2017 to February 2019) NCP before and after the cruise, biologically-induced changes in
DIC were estimated from the OOI OSP mooring. Water column DIC inventory (0-100 m) was obtained from the combination
of surface data from OSP mooring observations and subsurface data empirically estimated from the neural network algorithm
(CANYON-B algorithm, Bittig et al., 2018), which has been previously applied successfully to the OSP region (Haskell et al.,
2020; Huang et al., 2022). Depth-integrated (0 to 100 m) mooring-based NCP became positive (i.e., net autotrophic) around
February 2018 and 2019. Integrated NCP averaged $18.8 \pm 5.5$ mmol C m$^{-2}$ d$^{-1}$ between February and August 2018 (Fig. 3) and
averaged $11.0 \pm 0.8$ mmol C m$^{-2}$ d$^{-1}$ during the EXPORTS cruise period.

Comparing both mooring and glider estimates of integrated NCP (0-100 m) over the fully available time series presented here
(August to November 2018), we find that the glider estimates averaged +1.6 mmol C m$^{-2}$ d$^{-1}$ and the mooring estimates
averaged -4.5 mmol C m$^{-2}$ d$^{-1}$, which are both relatively small. These estimates suggest that NCP is reduced in the fall, which
aligns with previous seasonal NCP studies based at OSP (Fassbender et al., 2016; Haskell et al., 2020). However, there are
NCP offsets between the estimates during the EXPORTS cruise period (11.0 vs. 3.8 mmol C m$^{-2}$ d$^{-1}$ for the mooring vs glider).
Such offsets may be explained by the different temporal integration scales of the tracers considered (i.e., DIC vs. O$_2$), as has
been previously demonstrated using BGC float data from OSP (Yang et al., 2018; Huang et al., 2022). Given that oxygen-
based NCP measurements are of higher spatial and temporal coverage, we only compare glider-based integrated (0-100 m)
NCP during the cruise period. However, we use the mooring-based NCP estimates to demonstrate the potential contributions
of NCP between March and August.

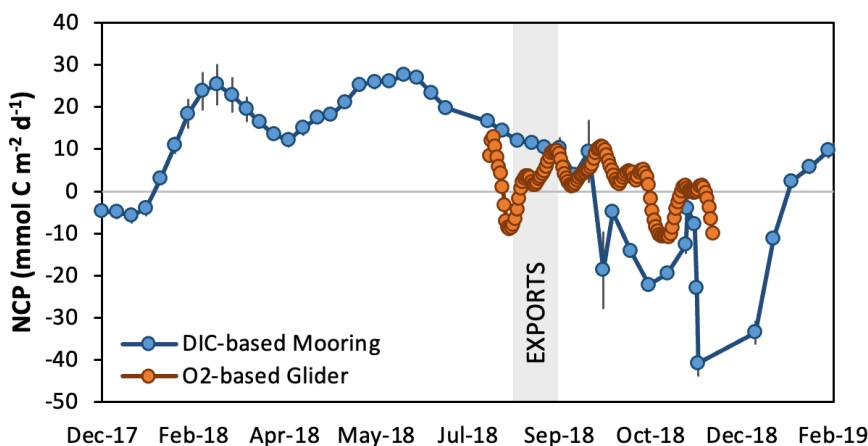

**Figure 3: Depth-integrated (0-100 m) net community production (NCP) based on surface Ocean Station Papa (OSP) mooring**
**measurements of dissolved inorganic carbon (December 2017 to February 2019) in blue and based on oxygen-sensor mounted to**
**gliders during site occupation (August to November 2018) in orange. The vertical grey bar denotes the EXPORTS field campaign**
**stationed near OSP.**





## 2.2 Carbon Demand in the Mesopelagic Zone

### 2.2.1 Bacterioplankton Respiration

Only respiration rates were used to measure carbon demand within the upper MZ (as opposed to respiration+production). Free-living bacterioplankton respiration rates were estimated using $^3$H-leucine incorporation rates ($^3$H-Leu incorporation, pmol L$^{-1}$ d$^{-1}$) measured over the upper 500 m. $^3$H-Leu incorporation rates were converted to daily bacterioplankton carbon production rates (BP) using a site-specific cell conversion factor $0.11 \cdot 10^{18}$ cells mol leucine$^{-1}$ (Kirchman, 1992) and a cruise-estimated relationship between cell C content and cell biovolume (fg C cell = $91.71 \cdot$(cell biovolume in $\mu m^3$)$^{0.686}$) (Stephens et al., 2023).

Bacterioplankton respiration (BR) was calculated from daily estimates of BP and cruise-estimated bacterioplankton growth efficiency (BGE) of 19% for depths between 95 and 500 m (Stephens et al., 2020) using the formula BR = BP/BGE. Between 95 and 500 m, cruise-mean BR rates decreased from $0.028 \pm 0.002$ mmol C m$^{-3}$ d$^{-1}$ at 95 m to $0.0079 \pm 0.0003$ mmol C m$^{-3}$ d$^{-1}$ at 195 and 330 m to $0.010 \pm 0.003$ mmol C m$^{-3}$ d$^{-1}$ at 500 m (Fig. 2j). BR rates significantly decreased (two-sample t-test, $p < 0.001$) with depth within the upper MZ, dropping by a factor of 2.5 between 95 and 150 m and by a factor of 1.3 between

150 and 195 m. Integrated BR (95-500 m) averaged $4.0 \pm 0.3$ mmol C m$^{-2}$ d$^{-1}$. Although cruise-averaged BR rates for Epochs 1 and 2 at 500 m were two-fold and significantly more than Epoch 3 (two-sample t-test, $p < 0.001$), there were no significant differences in the 95 to 500 m depth-integrated BR rates between Epochs (two-sample t-test, $p > 0.05$). Epoch 1, 2, and 3 rates averaged $3.9 \pm 0.4$, $4.2 \pm 0.9$, and $3.7 \pm 0.5$ mmol C m$^{-2}$ d$^{-1}$, respectively (error bars here represent propagated standard errors from triplicate measurements), exhibiting minimal changes in the profiles within the MZ between Epochs aside from the

increase at 500 m during Epochs 1 and 2.

Rates of particle-attached bacterioplankton respiration were estimated at three depths (105, 155, and 205 m) using oxygen-based removal rates on intercepted and incubated particles using the RESPIRE trap method (Boyd et al., 2015; Bressac et al., in review). Particle-attached respiration rates were $7.68 \pm 0.03$ mmol C m$^{-3}$ d$^{-1}$ at 105 m and $10.4 \pm 0.04$ mmol C m$^{-3}$ d$^{-1}$ at 205

m (Paul et al., in prep; Supp. Fig. 2). An increase in respiration was also observed when rates were normalized to the concentration of POC at each depth, resulting in normalized rates of $0.127$ d$^{-1}$ at 105 m and $0.285$ d$^{-1}$ at 205 m. Particle-bound respiration was 9%, 7%, and 17% of sinking particle flux at 105, 155 and 205 m, respectively. Applying a particle-bound respiration mean of 11% of sinking flux to all depths throughout the upper MZ, the cruise-mean depth-integrated (95 to 500 m) particle-bound bacteria respiration was estimated to be $0.5 \pm 0.2$ mmol C m$^{-2}$ d$^{-1}$.


### 2.2.2 Respiration by Bacterivorous Microzooplankton

Although respiration by microzooplankton (i.e., plankton < 20 μm) was not directly measured during the EXPORTS 2018 cruise, we use estimated grazing rates and carbon demand-based estimates of microzooplankton respiration based on experiments conducted during the cruise. Briefly, bacterioplankton grazing rates were calculated based on growth rates in dark





incubations conducted at 95 m (Stephens et al., 2020) by comparing growth rate differences between grazer-diluted and no

dilution conditions, and we assumed a decreased growth rate was due to microzooplankton grazing. Mean microzooplankton

abundances of $5 \cdot 10^4$ cells L$^{-1}$ and 20-40% assimilation efficiencies were tested (Straile, 1997; Landry and Calbet, 2004).

Depth-integrated (95 to 500 m) upper MZ microzooplankton respiration was estimated to be $0.4 \pm 0.2$ mmol C m$^{-2}$ d$^{-1}$. The

entirety of this carbon demand would be met in the upper MZ.


### 2.2.3 Mesozooplankton Respiration

Regionally validated allometric equations based on animal size (Maas et al., 2021b; Steinberg et al., 2023) were used to derive

respiration rates for net-collected resident mesozooplankton. Residents must meet their metabolic demands by carbon supplied

into the MZ, whereas migrators can consume (and respire) carbon inside and outside the MZ. Upper MZ-integrated (100 to

500 m) resident mesozooplankton respiration rates were relatively constant (Fig. 2k), with a cruise average of 0.7 mmol C m$^{-2}$ d$^{-1}$. After integrating over the upper MZ, residents contributed to the greatest fraction of mesozooplankton respiration,

representing 70-76% of the total mesozooplankton biomass. Including mesozooplankton migrators (Fig. 2l) would result in a

total mesozooplankton respiration rate of $0.8 \pm 0.1$ mmol C m$^{-2}$ d$^{-1}$.

### 2.2.4 Seasonal Respiration

Upper MZ respiration rates were derived from a BGC Argo float and a Seaglider deployed in August and July 2018. A time

rate of change of [O$_2$] from August 1$^{st}$ to November 1$^{st}$, 2018, was calculated by a linear fit to [O$_2$] measurements along

isopycnal surfaces and then gridded at 1 m depth resolution based on the average depth of isopycnals during the analysis

period. Noise due to water-mass variability was reduced by removing the detrended spice (i.e., density-compensated salinity

anomalies) versus [O$_2$] correlation. The primary assumption in this approach is that the observed [O$_2$] changes are a first-order

process due to biological activity (Hennon et al., 2016; Arteaga et al., 2019; Billheimer et al., 2021). As such, attenuation of

[O$_2$] was highest where the attenuation of sinking POC flux, DOC concentrations, and bacterioplankton biomass was also the

greatest between 100 and 200 m (Fig. 2i). Using Laws (1991) plankton biomass-based conversions to C units (O$_2$:C of 1.4:1),

the integrated respiration rate from the float data was $14.1 \pm 8.2$ mmol C m$^{-2}$ d$^{-1}$ and that from the glider data was $16.7 \pm 2.7$

mmol C m$^{-2}$ d$^{-1}$.

### 2.3 Supply and Demand Based on [POC] Data Inversion

In addition to the carbon supply and demand estimates described above, an inverse method was used to produce internally

consistent estimates of the rates of multiple particle cycling processes at OSP during the EXPORTS cruise. In this method,

[POC] data were used to estimate the rate parameters of a POC cycling model that includes POC production (NPP), passive



sinking, (dis)aggregation, vertical transport due to zooplankton diel vertical migration (DVM), and remineralization. Specifically, the method combined (i) measurements of size-fractionated (1-51 μm and > 51 μm) [POC] obtained from large-volume in situ filtration and (ii) a two-particle size-class model of POC cycling in the upper 500 m (Amaral et al., 2022). Based on this method, estimated cruise-mean POC fluxes decreased from 2.2 ± 0.8 mmol C m$^{-2}$ d$^{-1}$ at 100 m to 0.9 ± 0.3 mmol C m$^{-2}$ d$^{-1}$ at 500 m, resulting in a flux attenuation of 1.3 ± 0.9 mmol C m$^{-2}$ d$^{-1}$ that matches the average estimate derived from traps and radionuclides.

Most of the flux attenuation occurred in the 100-150 m layer (58%) and was primarily attributed to 1-51 μm particles (77%) rather than > 51 μm particles (Amaral et al. 2022). The vertical distribution of the estimated flux attenuation is also consistent with observations from traps and radionuclides. The cruise-mean NPP derived by extrapolating an exponential decrease in the EZ to the upper MZ was 0.4 ± 0.1 mmol C m$^{-2}$ d$^{-1}$ (85% in the 100-150 m layer, Fig. 2h). The estimated cruise-mean DVM-related supply of > 51 μm POC to the upper MZ was 2.0 ± 1.6 mmol C m$^{-2}$ d$^{-1}$, four-fold more than the cruise-based DVM measurements, but within one standard deviation. The cruise-mean POC remineralization in the 100-500 m layer was estimated to be 12.6 ± 2.9 mmol C m$^{-2}$ d$^{-1}$, more than double cruise-based respiration measures mentioned above but similar to the long-term oxygen trends. Note that in the study of Amaral et al. (2022), "POC remineralization" is defined as the loss of POC to DIC, DOC, or particulate size classes < 1 μm, so that it would be more accurately termed "POC remineralization and solubilization" and is expected to be larger than estimates of true respiration to DIC. Most of the so-called remineralization occurred between 200 and 500 m (61%) and corresponded to 1-51 μm particles (84%; Amaral et al., 2022). Because the POC budget must be closed in this model, a "residual" for the 100-500 m layer was estimated to be 8.8 ± 6.6 mmol C m$^{-2}$ d$^{-1}$. This residual may result from several factors, including a missing source of POC to the mesopelagic, physical transport, and/or unsteadiness (Amaral et al. 2022).

## 2.4 Comparison of Supply and Demand

The upper MZ (100 to 500 m) EXPORTS OSP organic carbon budget is summarized in Figure 4, which combines the depth-integrated supply and demand estimates. The organic carbon supply estimates into the upper MZ include NPP, carbon inputs from vertically migrating zooplankton, sinking particle flux attenuation, and DIC fixation via chemoautotrophy. Combined estimates resulted in a cruise-based total supply of 3.0 mmol C m$^{-2}$ d$^{-1}$. This estimate matches the NCP measured in the euphotic zone during the cruise after subtracting the mean particle sinking flux at 500 m (3.0 ± 0.6 mmol C m$^{-2}$ d$^{-1}$, i.e., representing the potential mesopelagic carbon supply from NCP). The glider-based estimate of a relatively low NCP is in general agreement with several other estimates of NCP collected during the cruise period (e.g., wirewalker and incubation-based estimates; Neibergall et al., 2023). The demand estimates include respiration from bacteria (free-living and particle-attached) and zooplankton (micro- and meso-), which resulted in a total demand of 5.7 mmol C m$^{-2}$ d$^{-1}$. Total demand exceeds total supply by 2.7 mmol C m$^{-2}$ d$^{-1}$, indicating that the supply terms measured during the cruise were insufficient to meet the demand



measured during the same period (Fig. 4). Long-term estimates of DOC supplied from earlier in the year and respiration based on autonomous sensors are also shown for comparison, demonstrating potential temporally integrated contributions to supply and demand, respectively.

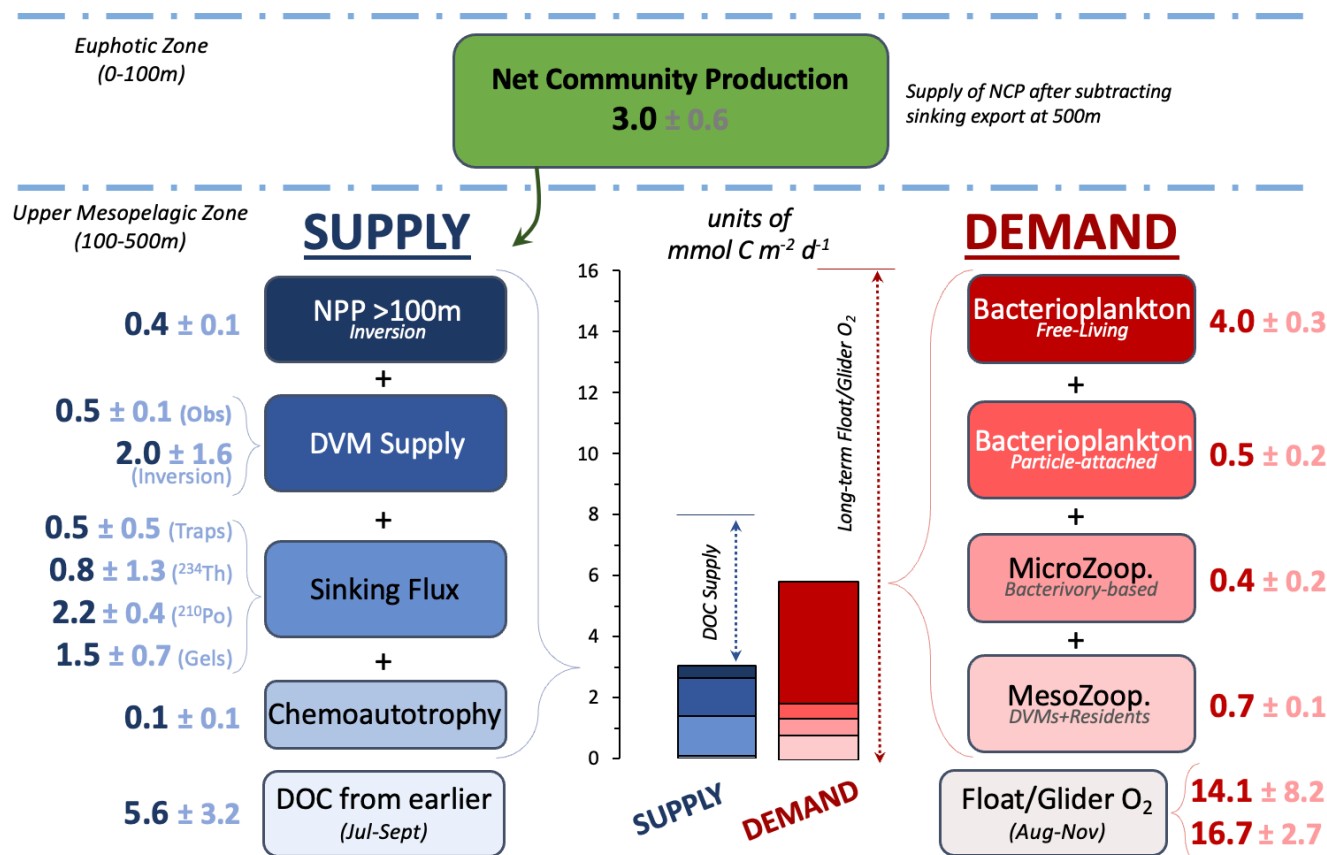

**Figure 4: Comparison of organic C supply and demand estimates for the upper mesopelagic zone (MZ; 100 to 500 m) and stacked**
**bar plot of combined supply and demand. Terms with multiple measurement methods (i.e., zooplankton diel vertical migration (DVM) supply, sinking flux attenuation, and float/glider respiration) were averaged for further carbon budget analysis. Cruise period-based net community production (NCP) is estimated from high-resolution glider $O_2$ observations and represents the difference after subtracting the mean sinking particle flux at 500 m (0.8 mmol C $m^{-2}$ $d^{-1}$). The dotted line above the supply stacked bar represents an additional supply term based on dissolved organic carbon (DOC) removal trends from July to September 2018**
**and the dotted line next to the demand stacked bar represents long-term respiration as a potential upper bound of respiration based on float and glider $O_2$ trends collected from August to November 2018.**

## 3 Discussion

The EXPORTS program study was designed to quantify a full range of carbon flux and attenuation pathways. Thus, the suite of observations collected during the North Pacific field campaign was perhaps the most comprehensive characterization of
organic carbon flux and attenuation into and through the upper mesopelagic zone (100-500 m) conducted at a single location





(Siegel et al., 2021). The multiple and independent measurements allowed for robust constraints on the uncertainties in the organic carbon budget for OSP's upper MZ.

## 3.1 Monte Carlo Constraint of Uncertainties

To best use the combination of measurements available for each process, we applied a Monte Carlo approach to assess overall uncertainty in carbon supply and demand terms. This approach was carried out by randomly assigning a value for each term drawn from a normal distribution with mean and uncertainty, as reported in Figure 4. When multiple approaches were used to measure the same quantity, we equally weighted the methods. For example, for each iteration of the DVM supply, we randomly pick from either the observation-based ($0.5 \pm 0.1$ mmol C m$^{-2}$ d$^{-1}$) or inversion-based values ($2.0 \pm 1.6$ mmol C m$^{-2}$ d$^{-1}$).

Probability distributions for supply and demand were generated (Fig. 5a) using a large number of randomly selected budget combination iterations (n = 10,000). Based on this uncertainty analysis, we estimate that carbon supply and demand were significantly different from one another ($3.0 \pm 1.7$ vs. $5.7 \pm 0.4$ mmol C m$^{-2}$ d$^{-1}$; t-test, $p < 0.01$; Fig. 5a) and that demand exceeded supply by $2.7 \pm 1.8$ mmol C m$^{-2}$ d$^{-1}$ (Fig. 5b). Demand exceeded supply in 92% of the Monte Carlo simulations, suggesting that there was a relatively low potential for supply to have met demand.


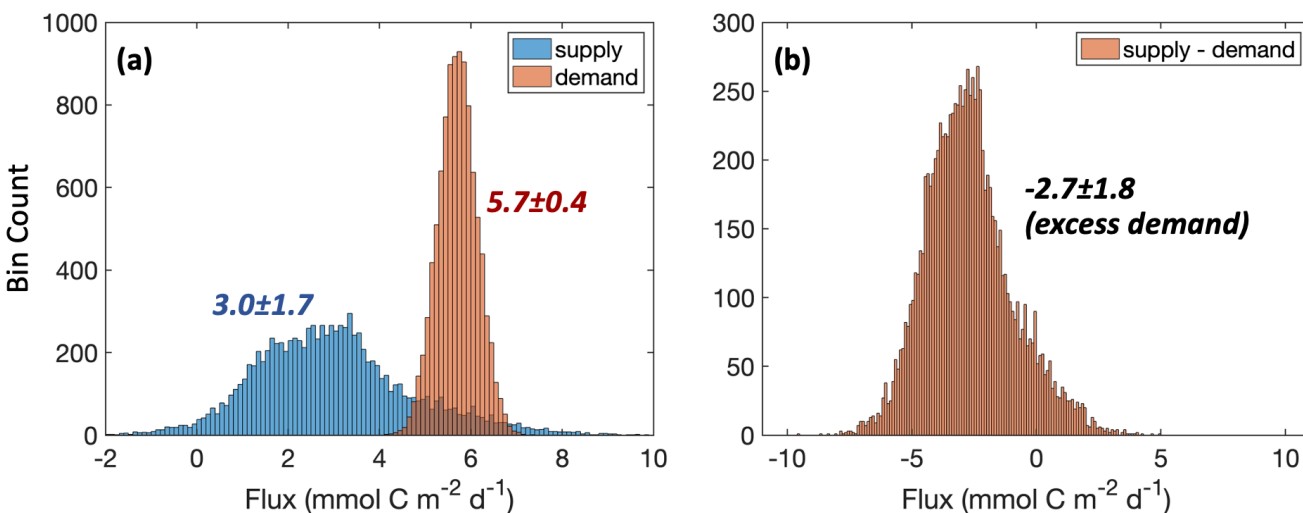

**Figure 5: Monte Carlo-based distribution of carbon supply and demand in the upper mesopelagic zone (100-500 m) (a) and the difference between supply and demand (b). Using the mean and uncertainty (i.e., standard deviation or standard error depending on the methodological approach) for each process shown in Figure 4, 10,000 random samplings of a mean ± uncertainty were**
**combined to provide a "grand distribution" of all respective supply and demand processes. As seen in the final grand means, this approach assumes the estimated mean and uncertainty represent normal distributions. Note that modeled data and literature-based supply estimates by chemoautotrophic production and demand by microzooplankton respiration were included in the analysis; however, cruise-based NCP, long-term DOC removal, and long-term respiration from the float/glider were not included.**



## 3.2 Resolving Supply and Demand Imbalance

Part of the excess demand not accounted for by organic carbon supplied during the cruise period (-2.7 ± 1.8 mmol C m$^{-2}$ d$^{-1}$) may have come from slowly sinking and slowly solubilized particles supplied from biological production earlier in the year (Fig. 6). According to mooring-based potential density profiles, the most recent mixing event at OSP was in March 2018 (Supp. Fig. 3). However, this mixing was not deeper than 100 m. Therefore, seasonal vertical mixing could not have delivered EZ organic carbon into the upper MZ. Instead, sinking particles from elevated springtime productivity could have been transformed into very slowly (e.g., <1 m d$^{-1}$) and non-sinking POC and DOC that were derived from NCP and exported earlier in the year. Spring and early summer NCP values in 2018 between 0 and 100 m were 22-25 mmol C m$^{-2}$ d$^{-1}$ (Fig. 3), as commonly observed for the region (Fassbender et al., 2016; Haskell et al., 2020). This surface elevation of NCP in spring and early summer at OSP has been associated with increased backscattering-based estimates of [POC] in subsurface depths (50 to 120 m; Huang et al., 2022). The input of slowly sinking particles to the MZ is also consistent with three independent findings from EXPORTS indicating (1) that particle disaggregation was a major loss process of large particles in the EZ and below (Amaral et al., 2022), (2) that most POC throughout the upper MZ had sinking velocities < 10 m d$^{-1}$ (Romanelli et al., 2023), and (3) that small particles in the upper MZ reflected a microbial particle solubilization signature that increased with depth (Wojtal et al., 2023).

Consumption of exported organic carbon from earlier in the year is indicated by the long-term removal of DOC, which decreased over three discrete time points between July and September (Supp. Fig. 1). Bulk DOC concentrations are comparatively large (42-58 µM DOC vs. 1-4 µM POC). Only 10's nM DOC d$^{-1}$ would need to be removed at given depths in the upper MZ to resolve the organic matter supply and demand mismatch. DOC decreased at all depths between July and September by ~ 1.1 µM C on average, which is close to our analytical ability to resolve change (i.e., doubling the CV of 1.3% results in uncertainty concentrations of ±1.2 to ±1.4 µM for these depths). While such small changes in DOC concentrations are not possible to resolve daily throughout a cruise due to methodological limitations, it is possible to determine if a change in DOC is monitored over timescales of weeks to months. However, this approach requires the assumption that the change in DOC is not impacted by mixing in the MZ and assumes lateral advection in the MZ is minimal. Mean lateral velocities for th1e upper MZ near OSP are 0.5-1.5 km d$^{-1}$ (Hristova et al., 2019), similar to observed float advection during 2018-2019 (Siegel et al., 2021; Huang et al., 2022). So, upper MZ waters would not have moved by more than 70 km between July and September, smaller than typical subarctic North Pacific mesoscale eddies (e.g., Cheng et al., 2014). If the upper MZ only received inputs from surface production (i.e., no lateral supply), which is supported by the slow transport times at these depths, DOC was most likely either supplied from solubilized particles that were exported into the upper MZ via NCP or released from zooplankton earlier in the year.



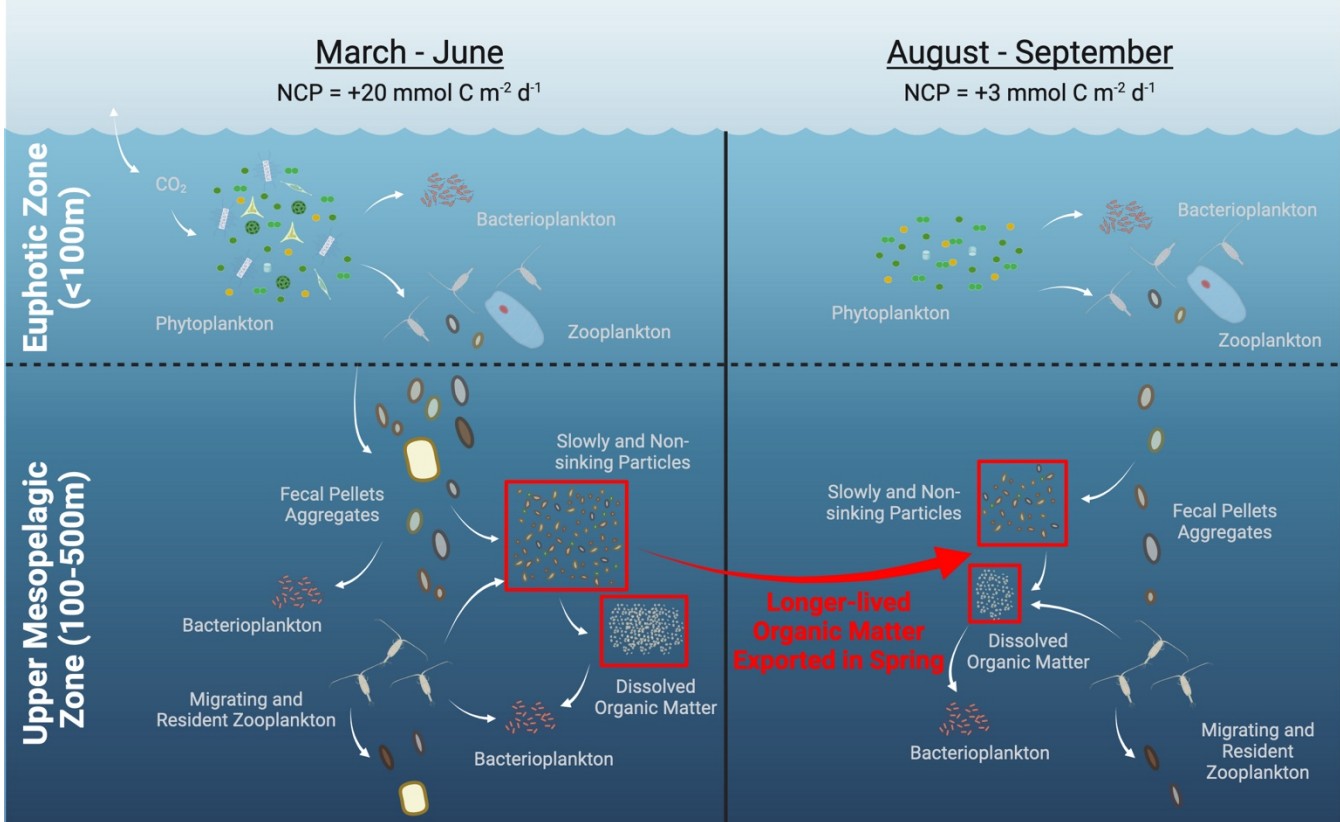

**Figure 6: High net community production (NCP) in spring at Ocean Station Papa may have led to accumulating longer-lived, slowly-
and non-sinking organic matter in the upper mesopelagic zone. Potential pathways for generating this accumulated organic matter
could include particle disaggregation, solubilization, or release from vertically migrating zooplankton. This accumulated non-
sinking organic matter may have contributed to the excess carbon demand in August 2018 (large red arrow) required for respiration
compared with instantaneous ship-based supply. Created with BioRender.com.**

The imbalance in supply and demand was somewhat unsurprising, given that prior studies at OSP had also concluded that
subsurface carbon demand likely exceeds concurrent carbon export measurements (Boyd et al., 1999). OSP is generally
considered a stable open ocean environment due to low eddy kinetic energy, high-nitrate, low-chlorophyll conditions, and
permanent stratification that typically limits convective mixing to 100 m (Whitney and Freeland, 1999; Harrison, 2002).
Characteristics of OSP include surface communities dominated by small phytoplankton and tightly coupled food webs (Boyd

and Harrison, 1999). Therefore, given that OSP fits the canonical mold of a balanced, slowly evolving system, we hypothesize
that the pronounced decoupling between carbon supply from the EZ and demand within the upper MZ may be a prevalent
feature of such environments. As previously hypothesized (Fassbender et al., 2016; Huang et al., 2022), we posit that a
seasonally active DOC pool must also be associated with production from earlier in the year to meet the estimated demand
from autonomous and in situ observations. In the absence of strong convective mixing, organic carbon may have been supplied



to the upper MZ earlier in the year by some combination of POC solubilization, as shown by backscattering data (Huang et al. 2022) and as release from zooplankton (Lopez et al., 2020).

### 3.3 Uncertainties and Literature Comparison

Several sources of uncertainty are associated with timescales, measurement errors, and underlying assumptions across the various carbon supply and demand terms, summarized in Table 1 and Appendix 2 and highlighted below. These sources can 480    inform future strategies to constrain marine organic carbon mass balance and monitoring efforts.

**Table 1. Estimates of integrated (either 0-100 m for NCP or 100-500 m for other estimates) supply and demand into the upper mesopelagic zone (100-500 m). Also noted are how the estimated uncertainty was calculated for each method,**
**the assumptions and limitations for each method, and associated references.**

| Type of Estimate | Method | Estimate (mmol C $m^{-2}$ $d^{-1}$) | Uncertainty Calculation | Assumptions/ Limitations | References |
|---|---|---|---|---|---|
| *Supply* | | | | | |
| NCP during cruise period (0-100 m) | Seagliders ($O_2$ mass balance) | 3.0±0.6 | Standard deviation of 1000 runs of an oxygen mass balance model | Assumption of horizontal advection and diffusion fluxes = 0 | Traylor et al., submitted |
| Inverse NPP >100 m | Size-fractionated POC inversion | 0.4±0.1 | Propagated from posterior estimates of POC concentrations and particle cycling parameters | Steady state, omission of horizontal and vertical physical transport | Amaral et al., 2022 |
| | | | | Possible misrepresentation of particle cycling processes in the model: posterior estimates of NPP agree well with NPP measurements in the euphotic zone, but model assumes an exponential decrease in NPP with depth, leading to relatively high integrated estimated NPP in the mesopelagic zone | |
| DVM supply | Allometric-based | 0.5±0.1 | Standard deviation of estimates of DVM flux from the 3 epochs. Each epoch was an average of two biological replicates | Net avoidance (including macrozooplankton, nekton, and fish) | Stamieszkin et al., 2021; Maas et al., 2021a; Steinberg et al., 2023 |
| | | | | Possible impact of zooplankton patchiness | |
| | | | | Application of allometric equations | |



| Type of Estimate | Method | Estimate (mmol C m$^{-2}$ d$^{-1}$) | Uncertainty Calculation | Assumptions/ Limitations | References |
|---|---|---|---|---|---|
| | Size-fractionated POC inversion | 2.0±1.6 | Propagated from posterior estimates of POC concentrations and particle cycling parameters | Steady state, omission of horizontal and vertical physical transport | Amaral et al., 2022 |
| | | | | Possible misrepresentation of particle cycling processes in the model: representation of DVM is quite simple, and may overpredict actual DVM | |
| Passive flux from sinking particles | Traps | 0.5±0.5 | Propagated using standard error of the mean of POC fluxes by epoch | Possible undersampling of large/rare and small/slowly-sinking particles | Estapa et al., 2021 |
| | | | | Possible impact of swimmers | |
| | Gel traps | 1.5±0.7 | Propagated using standard error of the mean of POC fluxes by epoch | Calculated/modeled values from images using best-fit parameters for volume and carbon content | Durkin et al., 2021 |
| | | | | Possible trap undersampling of particles | |
| | $^{234}$Th | 0.8±1.3 | Propagated using standard deviation of $^{234}$Th activities, $^{234}$Th fluxes, and POC/$^{234}$Th ratios | Consideration of two end-member scavenging models | Buesseler et al., 2020; Roca-Martí et al., 2021 |
| | | | | Assumption of POC/$^{234}$Th ratios in sinking particles based on size class | |
| | | | | Possible impact of DVM | |
| | $^{210}$Po | 2.2±0.4 | Propagated using standard error of the mean of POC fluxes by epoch | Assumption of a SS model neglecting physical processes | Roca-Martí and Puigcorbé, 2024 |
| | | | | Assumption of POC/$^{210}$Po ratios in sinking particles based on size class | |
| | Size-fractionated POC inversion | 1.4±0.9 | Propagated from posterior estimates of POC concentrations and particle cycling parameters | Steady state, omission of horizontal and vertical physical transport | Amaral et al., 2022 |
| | | | | Possible misrepresentation of particle cycling processes in the model | |
| Chemoautotrophic production | From field-based nitrification rates and | 0.1±0.1 | Fraction of nitrification and its associated error (Santoro pers. | DIC fixation is 1 tenth of nitrification | Bayer et al., 2023; Reinthaler et al., 2010 |
| | | | | Assumption of C:N ratio | |



| Type of Estimate | Method | Estimate (mmol C m$^{-2}$ d$^{-1}$) | Uncertainty Calculation | Assumptions/ Limitations | References |
|---|---|---|---|---|---|
| | literature-based conversions | | comm, manuscript in prep) | | |
| Long-term DOC removal | DOC measurements over time | 5.6±3.2 | Propagated error using standard deviation of triplicate measurements for each time point | The difference between measurements over time was often less than the mean instrumental uncertainty of 1.4 μM | This study |
| Residual supply | Size-fractionated POC inversion | 8.8±6.6 | Propagated from posterior estimates of POC concentrations and particle cycling parameters | Steady state, omission of horizontal and vertical physical transport<br><br>Possible misrepresentation of particle cycling processes in the model: POC budget must be perfectly satisfied; this residual term closes the budget | Amaral et al., 2022 |
| ***Demand*** | | | | | |
| Free-living bacterial respiration | $^3$H-leucine incorporation rates and bacterial growth efficiency estimates | 4.0±0.3 | Propagated using standard error of the mean of bacteria respiration by epoch | Uncertainties in carbon conversion and bacterial growth efficiency | Stephens et al., 2020 |
| Particle-attached bacterial respiration | RESPIRE traps | 0.5±0.2 | Propagated using error of linear regression fit to oxygen consumption data. Carbon-normalized respiration rates for RESPIRE traps had errors that were propagated also using standard error of the mean of POC and DOC concentrations | Potential artifacts of the RESPIRE trap method (Boyd et al., 2015)<br><br>Assumption of $O_2$:C ratio | Boyd et al., 2015 |
| Microzooplankton respiration | From literature | 0.4±0.2 | Propagated error from loss of bacteria in incubations from Stephens et al. (2020) | Assumption of assimilation efficiency of 20-40% from Straile (1997) | This study |
| Mesozooplankton respiration | Allometric-based | 0.8±0.1 | Standard deviation of estimates of zooplankton | Net avoidance (including macrozooplankton, nekton, and fish) | Maas et al., 2021a; |



| Type of Estimate | Method | Estimate (mmol C m$^{-2}$ d$^{-1}$) | Uncertainty Calculation | Assumptions/ Limitations | References |
|---|---|---|---|---|---|
| | | | respiration from the 3 epochs. Each epoch was an average of two biological replicates | Possible impact of zooplankton patchiness | Steinberg et al., 2023 |
| | | | | Application of allometric equations | |
| Transfer from POC to dissolved (<1 μm) phase | Size-fractionated POC inversion | 12.6±2.9 | Propagated from posterior estimates of POC concentrations and particle cycling parameters | Steady state, omission of horizontal and vertical physical transport | Amaral et al. 2022 |
| | | | | Possible misrepresentation of particle cycling processes in the model | |
| Long-term respiration | Seagliders (O$_2$ mass balance) BGC-float (O$_2$ mass balance) | 16.7±2.7 14.1±8.2 | Standard error of slope in linear regression | Assumption of O$_2$:C ratio, limited ability to account for lateral advection | This study |

## 3.4 Constraining Carbon Supply

### 3.4.1 Passive Flux

Several methods were used to constrain organic carbon sources to the upper MZ during the cruise period. Passive sources of

carbon supply were estimated via sediment traps (surface-tethered and neutrally-buoyant), two particle-reactive radiotracers ($^{234}$Th and $^{210}$Po), gel trap collectors, and size-fractionated POC data. The use of these five independent approaches led to cruise-mean flux attenuation estimates from 100 to 500 m that ranged by a factor of 4 (Fig. 4). Taken together, the cruise-average flux attenuation between 100 and 500 m at OSP was $1.3 \pm 0.6$ mmol C m$^{-2}$ d$^{-1}$, equivalent to 42% of the cruise-average supply and 22% of the cruise-average demand.


Differences between methods can stem from the uncertainties, limitations, and assumptions specific to each method (Table 1, Appendix 2). For example, the discrepancy between the two methods that resulted in the lowest and highest flux attenuation estimates, sediment traps ($0.5 \pm 0.5$ mmol C m$^{-2}$ d$^{-1}$) and $^{210}$Po ($2.2 \pm 0.4$ mmol C m$^{-2}$ d$^{-1}$), could be due to incorrect assumptions in the $^{210}$Po model (Roca-Martí and Puigcorbé, 2024), zooplankton active migrant fluxes not caught by traps or trap

undersampling of particles (Estapa et al., 2021). On the other hand, the temporal and spatial scale covered by each method may have also contributed to differences in attenuation estimates. For instance, sediment traps were deployed from the process ship for 3-6 days during each Epoch, whereas $^{210}$Po-derived fluxes were estimated from seawater and size-fractionated particle profiles collected from the survey ship once per Epoch. Indeed, POC flux temporal variability and/or spatial patchiness are suggested by flux attenuation estimates that varied by up to 1.0 mmol C m$^{-2}$ d$^{-1}$ for traps and 1.6 mmol C m$^{-2}$ d$^{-1}$ for $^{210}$Po.

Moreover, the difference in POC flux estimates from traps and $^{210}$Po may also be explained by the fact that trap estimates



represent snapshots of sinking particle flux during the trap deployment duration. In contrast, flux estimates from [210]Po were obtained by using a steady-state (SS) model and integrate sinking particle processes on timescales of ~80 days, i.e., from June to the EXPORTS sampling period (Buesseler et al., 2020; Turnewitsch et al., 2008). Therefore, the higher flux attenuation derived from [210]Po may reflect stronger attenuation over the months before sampling in the study area.


Further, POC fluxes estimated at the upper boundary of the upper MZ (100 m) using the long-term estimates from [210]Po (2.6 $\pm$ 0.3 mmol C m$^{-2}$ d$^{-1}$) are similar in magnitude to the flux estimates obtained using the other passive sinking flux methods (from 1.4 $\pm$ 0.5 to 2.2 $\pm$ 0.7 mmol C m$^{-2}$ d$^{-1}$), suggesting that input of sinking particles to the upper MZ was similar in magnitude between June and the EXPORTS occupation in August-September. This mismatch of MZ supply to demand points to the possibility that greater particle flux (and subsequent disaggregation and solubilization) may have occurred before June, possibly between March and May, in line with elevated NCP during that period in 2018 (Fig. 3). The potential for elevated export earlier in the year is also in agreement with the observed maximum subsurface backscattering-based estimate of [POC] at 100 to 120 m in March to April in 2019 (Huang et al., 2022). Additional datasets that integrate over longer timescales, like [210]Po estimates, or that can capture long-term carbon stock trends, like time-series mooring datasets or DOC removal, may help identify temporal decoupling between instantaneous ship-based carbon supply and demand measurements.

### 3.4.2 Active Flux

An important finding from this study is that the DVM supply significantly contributed to the total supply of organic carbon to the upper MZ. In our study, we used two approaches to estimate this supply term: the first was an inverse estimate, base don size-fractionated [POC] data, of POC supply to the upper MZ by DVMs, while the second used a combination of net tows, physiological experiments, and allometric modeling. Combining both methods, the cruise-average active flux from migrating zooplankton was 1.2 $\pm$ 1.3 mmol C m$^{-2}$ d$^{-1}$, equalling the contribution from passive flux attenuation to upper MZ carbon supply at OSP. Yet, surface-derived carbon carried to midwater depths by migrating organisms is one of the more difficult terms to estimate. As a result, supply from vertically migrating zooplankton, which includes DOC excretion, fecal pellet production (sometimes referred to as "gut load" from surface feeding), and the loss of zooplankton biomass to predation at depth, is rarely, or only partially, constrained in organic carbon budgets. Prior studies estimated DVM supply from DOC excretion and found that it was around 10% of passive flux supply in MZ carbon budgets in the North Pacific (ALOHA, K2; Steinberg et al., 2008) and North Atlantic (BATS, PAP; Steinberg et al., 2000; Giering et al., 2014). However, our observations suggest that mortality and fecal pellet egestion can increase DVM supply by a factor of 2.5 relative to estimates based only on DOC excretion.

The POC-derived estimate of POC supply from DVM (Amaral et al., 2022) assumed that migrating zooplankton grazed small particles in the euphotic zone at a rate consistent with a global analysis of mesozooplankton ingestion rates (Calbet, 2001) and that 70% of ingested POC was used for growth and metabolism (Steinberg and Landry, 2017), with the remaining 30% egested into the mesopelagic zone as large (>51 μm) POC (e.g., fecal pellets). The [POC]-derived estimate is four times higher than



can be reconciled with net-based approaches, and that does not include the contribution of DOC release by mesozooplankton.
The high estimate inferred from [POC] data may be partly due to the implicit assumption that all mesozooplankton graze in the EZ are vertical migrators (vs. the $50 \pm 16\%$ we observed). Such an assumption would overestimate how much egested POC is delivered to the upper MZ. In addition, net tows do not reliably capture larger organisms such as macrozooplankton, nekton (e.g., salps), and fish due to net avoidance (Saba et al., 2021). Thus, any contributions from these organisms to active POC flux were excluded from the net-based active flux calculation. Prior work on the morphology and quantity of fecal pellets
produced in the euphotic zone noted that there was a substantial contribution of larger macrozooplankton and nekton, which resulted in an 11.5-fold mismatch between sampled mesozooplankton community production and total metazoan fecal pellet carbon estimates (Durkin et al., 2021; Stamieszkin et al., 2021). The discrepancies between these two active flux measurements can also be due to these larger-size class organisms.

Despite the limitations of net sampling, the ability to model different organic carbon sources supplied by zooplankton, including DOC, POC, and direct biomass loss, provides an essential assessment of how different carbon pools are transported to the midwater. For instance, the allometric relationships indicate that POC can be a greater proportion of active flux for salps and larger organisms. DOC release remains relatively constant, and predation is a proportionally smaller fraction. Since each source fuels a different mesopelagic community (particle-attached bacterioplankton, free-living bacterioplankton, and resident
zooplankton, respectively), it is worth distinguishing among the source types. Calculations of these allometrically-based rates are becoming more common (Maas et al., 2021a; Kwong and Pakhomov, 2017; Kiko et al., 2020; Davison et al., 2013) but could benefit from improved validation of the relevant physiological rate processes in different size classes, taxonomic groups, and regions.

Previous EXPORTS studies found that salps, when present, had an outsized role in POC fluxes at OSP, given their ability to accumulate biomass quickly and produce large amounts of fast-sinking fecal pellets (Durkin et al., 2021; Stamieszkin et al., 2021; Steinberg et al., 2023). Yet, because of their ephemeral and patchy nature, their biomass and contribution to POC fluxes varied strongly across sampling periods. Salp pellet POC export was equivalent to 0-48% of total sinking POC at 100 m during Epoch 1, with an increased contribution of up to 57% of the total POC flux by 500 m (Durkin et al., 2021; Steinberg et al.,
2023). These observations suggested little attenuation of salp fecal pellets with depth, given their fast-sinking speeds and low microbial respiration rates. Still, little attenuation could also reflect salp DVM activity from the upper 100 m at night to 300-750 m during the day (Steinberg et al., 2023). These DVM patterns combined with little attenuation of salp pellets and carcasses can help explain the relatively small contribution of salps to total active flux attenuation found in this study for the upper MZ (100-500 m).


Aside from DVMs, contributions by upper trophic levels (e.g., fishes) can also be an important supply source into the dark ocean (Saba et al., 2021). For instance, prior estimates for the eastern North Pacific suggest that fish-mediated export (from



migratory and non-migratory fishes) represents < 10% to ~40% of the total carbon export, with the higher values more typical
of the oligotrophic North Pacific Subtropical Gyre (Davison et al., 2013). However, findings from Davison et al. (2013) suggest

that fish-mediated export likely contributes to export below 500 m (rather than within the 100 to 500 m range) and, therefore,
does not contribute a measurable supply term within the upper MZ as defined in the current study. As such, larger trophic
levels, like fish-mediated export, are not included in the carbon budget presented here.

## 3.5 Constraining Carbon Demand

### 3.5.1 Bacterioplankton Respiration

Similar to a previous study at OSP (Boyd et al., 1999), the largest contribution to upper MZ carbon demand was from free-
living bacterioplankton respiration (>70% of total respiration). Several uncertainties can affect the estimates of the
instantaneous rates of BR. The method presented here derives BR by dividing measures of BP by estimates of BGE. The
estimated BGE of 19% for the upper MZ in the current study is significantly greater than the median values from the open

ocean MZ of 8% (4-12% range; Giering et al., 2014; Baumas et al., 2023). Stephens et al. (2020) experimentally determined
BGE during the 2018 Pacific EXPORTS campaign for various depths by tracking changes in bacterioplankton biomass and
organic carbon concentrations in controlled dark incubations. Had more common MZ literature-based estimates been used
(e.g., BGE of 10%) in the current study, the integrated BR would have been twice the values presented here (Fig. 4; 4.0 mmol
C m$^{-2}$ d$^{-1}$ for 19% BGE vs. 8.0 mmol C m$^{-2}$ d$^{-1}$ for 10% BGE). Therefore, literature-based estimates would have further

exacerbated the difference between the upper MZ's cruise-based supply and demand estimates. However, previous studies
have argued that bacterioplankton biomass-based approaches like those used here may more accurately reflect true BGEs, as
opposed to those that use a combination of oxygen-based BR and $^3$H-based BP (Briand et al., 2004), given that there are fewer
carbon conversion assumptions in the biomass approach.

The $^3$H-Leu incorporation assay for BP also carries additional uncertainties related to incubation pressure and isotope and
carbon conversion factors. For instance, deck-based BP methods, like those employed during the EXPORTS field study, may
lead to under- (e.g., Tamburini et al., 2003) or over-estimates (e.g., Amano et al., 2023) when compared to rates measured at
in situ pressures. Additionally, cell carbon conversions presented here resulted in estimates of nearly half of those from prior
studies at OSP. For instance, direct estimates of cell biovolume suggest that cells had an average carbon content of 11 fg C

cell$^{-1}$ (Stephens et al., 2020) versus larger values of 20 fg C cell$^{-1}$ (Kirchman et al., 1993; Simon et al., 1992; Sherry et al.,
1999), which would have increased the BP (and therefore the BR) estimate by ~2-fold. Similarly, the empirically derived $^3$H-
Leu conversion factors for BP at OSP (Kirchman, 1992) used in the current study (0.11·10$^{18}$ cells mol Leu$^{-1}$ coupled with a
site-specific cell biovolume to carbon content conversion) were, on average, 1.4-fold lower than if using the more commonly
applied carbon conversion factor (e.g., 1.5 kg C pmol Leu$^{-1}$; Simon and Azam, 1989). The sensitivity of BR to these conversion



factors has been identified previously (Ducklow and Hill, 1985; Ducklow and Carlson, 1992; Giering and Evans, 2022), demonstrating that site-specific conversion factors are important in carbon flux comparisons. Based on the above factors, we argue that the site-specific conversion factors used here produce more realistic estimates than those using literature-based conversions, bringing our integrated cruise-based demand closer to the integrated cruise-based supply values for the upper MZ.

**3.5.2 Long-term Respiration from Autonomous Platforms**

Autonomous platforms such as floats, moorings, and gliders can provide a unique view of upper MZ carbon budgets as they provide spatial and temporal coverages that are not possible through other means. Since early deployments of oxygen sensor-equipped floats, seasonal oxygen concentration decreases in the MZ have been used to estimate respiration rates (e.g., Martz et al. 2008). The approach depends on the assumption that the time rate of change of $[O_2]$ is due primarily to respiration while

ignoring physical processes. This assumption works best in regions with limited horizontal advection/gradients and where $[O_2]$ is replenished seasonally by deep mixing (Palevsky and Nicholson, 2018). A further limitation is that a sufficiently long observational period is needed to quantify a robust trend, usually requiring a single linear trend fit to several months of observations. The selected respiration quotient ($\Delta O_2:\Delta C$) is another source of uncertainty, based primarily on the complete oxidation of particulate organic matter (Laws, 1991). Dissolved oxygen only registers full respiratory remineralization and

cannot account for transformations between POC and DOC pools or partial oxidation of organic matter.

# 4 Conclusions and Implications

The mesopelagic carbon budget presented here is based on an extraordinary suite of simultaneous measurements collected near Ocean Station Papa (OSP) during August 2018 using a Lagrangian-based framework complemented with long-term observations from autonomous platforms. OSP was chosen as a quiescent endmember site with low biological carbon pump

efficiency given its typical late-summer conditions (e.g., shallow mixed layer, small phytoplankton concentrations, tightly coupled food web) and low mesoscale kinetic energy (Siegel et al., 2016, 2021). Yet despite the relatively stable physical conditions at OSP during late summer, this study has found a mismatch, by a factor of two, between total supply and total demand of organic C in the upper MZ (100-500 m) during the EXPORTS shipboard sampling at the site.

The sampling strategy adopted during EXPORTS allowed the constraint of carbon supply and demand terms in the upper MZ using a wide range of methods, including several to quantify the same component of the budget (sinking flux attenuation, zooplankton diel vertical migration (DVM) supply, and long-term respiration). This multi-pronged approach combined with an uncertainty (Monte Carlo) analysis is an important step to overcome method-specific uncertainties and limitations, building confidence in our ability to constrain the budget.




During the EXPORTS cruise, the total supply to the upper mesopelagic ($3.0 \pm 1.7$ mmol C m$^{-2}$ d$^{-1}$) matched the potential carbon supply from net community production (NCP), with a similar contribution from sinking flux attenuation and DVM. However, this supply did not meet the upper mesopelagic demand ($5.7 \pm 0.4$ mmol C m$^{-2}$ d$^{-1}$). We suggest that this mismatch is not indicative of an underestimation of the supply or an overestimation of the demand but instead challenges the steady-

state assumption, i.e., that supply and demand measured during a given period (in this case, 27 days) must be the same. Elevated NCP observed at the site in spring and early summer relative to late summer and other evidence (e.g., long-term DOC removal) suggest that earlier production at OSP fuelled an important part of the demand measured during the EXPORTS shipboard sampling. On the other hand, long-term (August-November) respiration estimates from autonomous platforms suggest that total mesopelagic respiration increased after the field campaign. These findings stress the importance of long-term and

sustained observations of the biological carbon pump.

Below the euphotic zone down to 500 m is the region where the attenuation of POC flux is largest and differs the most between oceanic regions (Buesseler and Boyd, 2009). Based on our transfer efficiency estimates, more than half of the POC exported into the upper MZ is consumed within the upper MZ at OSP. Predicting and quantitatively constraining this attenuation and

the pathways of carbon flux out of the upper MZ are increasingly important efforts as we seek to establish carbon budgets at depths where it is sequestered away from the atmosphere. In the context of establishing monitoring, reporting, and verification (MRV) standards for carbon dioxide removal (CDR) efforts, we find that individual methods likely misrepresent the true uncertainty associated with quantifying carbon flux. In the current study, the range of methods employed allowed us to account for the uncertainty of each measurement, demonstrating with a high level of confidence that estimates of supply and demand

in this low flux system during the cruise period were not equal. Similar levels of confidence would require similarly diverse measurement approaches. Additionally, this project emphasizes that a full view of carbon accounting could only be achieved in our study region by considering a full range of carbon export pathways, including DVM.

Our synthesis efforts provide a clear set of recommendations for improved quantification of the biological carbon pump: (1)

Models and observations need to include relevant midwater processes (e.g., DVM, heterotrophic bacteria, larger trophic levels), as well as their uncertainties; (2) Observations must capture the range of time scales appropriate for assessing the contributions of various MZ supply and demand processes; and (3) Redundancy in methodology should be encouraged as it is required to reduce the bounds of uncertainties.

Adopting these recommendations requires new approaches and tools. Most of the measurement methods employed in EXPORTS were ship-based, and many could not be implemented with remote or autonomous sensors. Both aspects encourage a continued commitment to sustained process studies where measurements are conducted with autonomous systems and platforms, with periodic visits from research vessels. To achieve sufficient spatial and temporal coverages, we must develop



novel sensors, modeling efforts, or experimental methods to provide insight into subsurface processes on scales that are large
enough to inform both global biogeochemical models and the nascent marine CDR community.

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

# Appendix

### Appendix 1. Values used for Figure 2

Epoch and cruise integrated average supply and demand in the upper mesopelagic zone (100-500 m). E1 = Epoch 1, E2 = Epoch 2 and E3 = Epoch 3. N.D. = no data.

| Method | Depth (m) | POC flux (mmol C m⁻² d⁻¹) E1 | E2 | E3 | Cruise Avg |
|---|---|---|---|---|---|
| POC (Traps) Flux | 95-105 | 1.1±0.2 | 0.8±0.4 | 2.3±0.6 | 1.4±0.5 |
| | 145-155 | 0.9±0.3 | 0.8±0.1 | 1.4±0.4 | 1.0±0.2 |
| | 195-205 | 0.7±0.2 | 0.7±0.2 | 0.9±0.4 | 0.8±0.1 |
| | 330-340 | 0.7±0.1 | 0.5±0.1 | 0.8±0.5 | 0.7±0.1 |
| | 500-510 | 0.4±0.1 | 1.0±0.2 | 1.2±0.7 | 0.9±0.2 |
| POC (Gel Trap) Flux | 95-105 | 2.7±0.7 | 0.8±0.0 | 3.0±0.3 | 2.2±0.7 |
| | 145-155 | 0.3±0.0 | 0.5±0.1 | 0.9±0.1 | 0.6±0.2 |
| | 195-205 | 2.0±0.9 | 0.3±0.0 | 0.3±0.0 | 0.9±0.6 |
| | 330-340 | 1.7±0.6 | 0.2±0.0 | 0.7±0.3 | 0.9±0.4 |
| | 500-510 | 0.9±0.3 | 0.4±0.1 | 0.8±0.3 | 0.7±0.2 |
| POC (²³⁴Th) Flux | 100 | | | | 2.0±0.6 |
| | 150 | | | | 1.3±0.6 |
| | 200 | | | | 1.1±0.7 |
| | 330 | | | | 1.3±1.0 |
| | 500 | | | | 1.2±1.2 |
| POC (²¹⁰Po) Flux | 100 | 2.2±0.4 | 3.2±0.6 | 2.3±0.3 | 2.6±0.3 |
| | 150 | 0.9±0.2 | 1.3±0.3 | 1.8±0.4 | 1.3±0.3 |
| | 200 | 0.7±0.3 | 0.8±0.2 | 0.6±0.4 | 0.7±0.1 |
| | 330 | 1.1±0.5 | 0.6±0.4 | N.D. | 0.9±0.3 |



| | 500 | 0.7±0.4 | 0.2±0.2 | 0.2±0.3 | 0.4±0.2 |
|---|---|---|---|---|---|
| POC (Inversion) Flux | 100 | | | | 2.2±0.8 |
| | 150 | | | | 1.4±0.5 |
| | 200 | | | | 1.5±0.5 |
| | 330 | | | | 1.2±0.4 |
| | 500 | | | | 0.9±0.3 |

| | | Zooplankton Supply (mmol C m$^{-2}$ d$^{-1}$) | | | |
|---|---|---|---|---|---|
| Method | Depth (m) | E1 | E2 | E3 | Cruise Avg |
| DVM Mortality | 95-145 | 0.033±0.009 | 0.000±0.000 | 0.001±0.002 | 0.011±0.017 |
| | 145-195 | 0.015±0.021 | 0.048±0.035 | 0.062±0.059 | 0.042±0.039 |
| | 195-300 | 0.038±0.051 | 0.053±0.020 | 0.005±0.000 | 0.032±0.033 |
| | 300-400 | 0.115±0.131 | 0.050±0.033 | 0.083±0.012 | 0.083±0.067 |
| | 400-500 | 0.041±0.013 | 0.046±0.033 | 0.074±0.028 | 0.054±0.026 |
| DVM DOC Excretion | 95-145 | 0.034±0.005 | 0.000±0.000 | 0.002±0.003 | 0.012±0.017 |
| | 145-195 | 0.017±0.024 | 0.049±0.030 | 0.062±0.056 | 0.043±0.037 |
| | 195-300 | 0.040±0.053 | 0.050±0.025 | 0.005±0.002 | 0.032±0.034 |
| | 300-400 | 0.105±0.110 | 0.055±0.030 | 0.095±0.008 | 0.085±0.056 |
| | 400-500 | 0.033±0.010 | 0.039±0.029 | 0.071±0.028 | 0.048±0.026 |
| DVM POC Excretion | 95-145 | 0.008±0.008 | 0.000±0.000 | 0.000±0.000 | 0.003±0.004 |
| | 145-195 | 0.003±0.003 | 0.012±0.012 | 0.016±0.016 | 0.010±0.010 |
| | 195-300 | 0.009±0.009 | 0.013±0.013 | 0.001±0.001 | 0.008±0.008 |
| | 300-400 | 0.029±0.029 | 0.012±0.012 | 0.019±0.019 | 0.020±0.018 |
| | 400-500 | 0.011±0.011 | 0.012±0.012 | 0.018±0.018 | 0.014±0.006 |

| | | Zooplankton Respiration (mmol C m$^{-3}$ d$^{-1}$) | | | |
|---|---|---|---|---|---|
| Method | Depth (m) | E1 | E2 | E3 | Cruise Avg |
| Resident Respiration | 95-145 | 0.001±0.000 | 0.001±0.000 | 0.001±0.001 | 0.001±0.000 |
| | 145-195 | 0.001±0.000 | 0.002±0.001 | 0.002±0.001 | 0.002±0.001 |
| | 195-300 | 0.001±0.001 | 0.001±0.001 | 0.001±0.000 | 0.001±0.000 |




| | 300-400 | 0.003±0.001 | 0.002±0.000 | 0.002±0.001 | 0.002±0.001 |
| | 400-500 | 0.001±0.000 | 0.001±0.001 | 0.002±0.001 | 0.001±0.000 |
| DVM Respiration | 95-145 | 0.001±0.000 | 0.000±0.000 | 0.000±0.000 | 0.000±0.000 |
| | 145-195 | 0.000±0.000 | 0.001±0.001 | 0.001±0.001 | 0.001±0.001 |
| | 195-300 | 0.000±0.000 | 0.000±0.000 | 0.000±0.000 | 0.000±0.000 |
| | 300-400 | 0.001±0.001 | 0.000±0.000 | 0.001±0.000 | 0.001±0.001 |
| | 400-500 | 0.000±0.000 | 0.000±0.000 | 0.001±0.000 | 0.000±0.000 |

**POC inversion-based NPP in MZ (mmol C m$^{-2}$ d$^{-1}$)**

| Method | Depth (m) | Cruise Average |
|---|---|---|
| Size-fractionated POC inversion | 100-150 | 0.34±0.11 |
| | 150-200 | 0.05±0.03 |
| | 200-330 | 0.01±0.01 |
| | 330-500 | 0.00±0.00 |

**Community Respiration (mmol C m$^{-2}$ d$^{-1}$)**

| Method | Depth (m) | Aug-Nov Avg |
|---|---|---|
| Float (O$_2$) | 95-145 | 0.071±0.040 |
| | 145-195 | 0.059±0.027 |
| | 195-330 | 0.034±0.025 |
| | 330-500 | 0.018±0.008 |
| Glider (O$_2$) | 95-145 | 0.061±0.018 |
| | 145-195 | 0.071±0.008 |
| | 195-330 | 0.037±0.007 |
| | 330-500 | 0.030±0.003 |

**Bact. Respiration (mmol C m$^{-3}$ d$^{-1}$)**

| Method | Depth (m) | E1 | E2 | E3 | Cruise Avg |
|---|---|---|---|---|---|
| Bacterial Respiration | 95-145 | 0.024±0.002 | 0.028±0.003 | 0.030±0.003 | 0.028±0.002 |
| | 145-195 | 0.010±0.001 | 0.013±0.003 | 0.010±0.000 | 0.011±0.001 |
| | 195-300 | 0.007±0.000 | 0.008±0.001 | 0.009±0.001 | 0.008±0.000 |
| | 300-400 | 0.006±0.001 | 0.008±0.003 | 0.009±0.001 | 0.008±0.001 |



| 400-500 | 0.015±0.001 | 0.012±0.002 | 0.006±0.002 | 0.011±0.003 |


## Appendix 2. Uncertainties

### Carbon Supply to the Mesopelagic Zone

**Passive Flux from Sinking Particles**

**POC Flux from Traps**

A detailed discussion of the difficult-to-quantify sources of uncertainty in the sediment trap particulate organic carbon (POC) fluxes may be found in Estapa et al. (2021). Fluxes of [234]Th to the sediment traps were consistently lower than the fluxes predicted from the water column [234]Th activities. One cause may have been the flux carried by large, rare particles (e.g., salp

fecal pellets) that were not well-sampled during short deployments of only a few days, which would bias the trap POC fluxes low. It is also possible that fluxes of small, slowly sinking particles (which were responsible for more of the [234]Th flux) were not efficiently retained in the traps due to hydrodynamic biases. However, the surface-tethered and neutrally buoyant traps collected similar [234]Th fluxes. The sediment traps were also strongly affected by the intrusion of active zooplankton swimmers whose biomass could not be entirely removed during the shipboard processing of the samples. While a correction was applied

to compensate for this, the "swimmer corrected POC fluxes" may be biased low because the correction was based on ratios of POC, [234]Th, biogenic silica (bSi), and projected area of passively sinking particles that were measured primarily in the deeper samples.

**POC Flux from [234]Th**

A detailed description of the uncertainties associated with the cruise-average POC flux estimates from [234]Th can be found in Buesseler et al. (2020). Briefly, uncertainties were calculated by propagating the standard deviation of [234]Th fluxes across all the stations (steady-state, SS), the standard deviation of depth-averaged [234]Th activities in Epochs 1 and 3 (non-steady-state, NSS), and the standard deviation of POC/[234]Th ratios in 5-51 μm particles across the pump stations. The uncertainty of the cruise-average POC flux attenuation reported here (0.8 ± 1.3 mmol C m$^{-2}$ d$^{-1}$) was calculated by propagating the uncertainties

of the cruise-average POC fluxes at 100 m (2.0 ± 0.6 mmol C m$^{-2}$ d$^{-1}$) and 500 m (1.2 ± 1.2 mmol C m$^{-2}$ d$^{-1}$). Below is a list of potential sources of uncertainty that should also be considered:

**The scavenging model choice (SS vs. NSS)**

Buesseler et al. (2020) present a two-end-member analysis of flux considering an SS and an NSS model to calculate cruise-

average POC fluxes for EXPORTS 2018. If only the SS or the NSS model were used, POC flux attenuation in the mesopelagic



zone would be $0.7 \pm 2.3$ (NSS) or $1.0 \pm 1.2$ (SS) mmol C m$^{-2}$ d$^{-1}$. Using an NSS model instead of an SS model is valid when there is significant variability in export within the timeframe of the $^{234}$Th half-life (24.1 d). The SS assumption may underestimate $^{234}$Th removal when there is intense scavenging and may overestimate $^{234}$Th removal during post-bloom conditions (Buesseler et al. 1992; Ceballos-Romero et al. 2018).


**Impact of physical transport**

Although we cannot separate temporal and physical processes, Buesseler et al. (2020) quantified the potential effect of physical transport, assuming that the increase in $^{234}$Th activity over time was solely due to physical processes. This analysis suggested that horizontal and vertical transport could reduce SS-derived $^{234}$Th fluxes in the surface by approximately 20%, resulting in

POC flux attenuation in the mesopelagic of up to $1.0 \pm 1.2$ mmol C m$^{-2}$ d$^{-1}$.

**Cruise-average POC flux vs. Epoch-average POC flux**

The choice to use a cruise-average POC flux versus an Epoch-average POC flux can also bias data. In this study, POC flux attenuation could increase to $1.1 \pm 0.2$ mmol C m$^{-2}$ d$^{-1}$ if Epoch-average POC fluxes were used (only SS $^{234}$Th model) instead

of an overall cruise average (considering the SS and the NSS $^{234}$Th models).

**POC/$^{234}$Th ratio choice (particle size class)**

The choice of particle size class as representative of sinking particles can greatly influence calculated POC fluxes and attenuation (Buesseler et al., 2020; Roca-Martí et al., 2021). In this case, POC flux attenuation could be between the range of

$0.2 \pm 2.6$ (>51 μm size class) and $2.4 \pm 1.4$ (1-5 μm size class) mmol C m$^{-2}$ d$^{-1}$, depending on the size class chosen. While the potential sources of uncertainty stated above would only change flux attenuation by 20-30%, the POC/$^{234}$Th ratio choice has the largest potential impact on results (factor of 3-5 difference). However, all potential flux attenuation estimates overlap with uncertainties with the estimate reported in this study. Together, this suggests that flux attenuation from 100-500 m was <2.5 mmol C m$^{-2}$ d$^{-1}$.


**Impact of DVM**

A potential bias in $^{234}$Th-derived fluxes that have not traditionally been examined is the impact of diel vertical migration (DVM). $^{234}$Th-derived fluxes should capture any mechanism that removes $^{234}$Th from seawater, including the gravitational sinking of particles (passively sinking particles) and the active transport of surface-derived particles to depth via DVM. In

contrast, particle fluxes measured using sediment traps would only (or mostly) reflect sinking particles. Differences in $^{234}$Th fluxes derived from seawater (predicted $^{234}$Th flux) and those concurrently measured with sediment traps have typically been used to measure trap collection efficiencies, with reported differences going in both directions (Buesseler et al., 2007). If DVM were a large factor in $^{234}$Th flux, the predicted $^{234}$Th flux would always be greater than the measured $^{234}$Th flux. In EXPORTS 2018, $^{234}$Th-derived water column fluxes were higher than those measured by traps. If DVM is significant, this difference



could reflect both trap undersampling biases and fluxes by DVM zooplankton that bypassed the sediment traps (Estapa et al., 2021). For the purposes of this study, if we assume that the difference in POC fluxes between sediment traps and water column-derived $^{234}$Th fluxes solely reflects DVM, we estimate that around 70% of the $^{234}$Th-derived POC fluxes at 100 and 500 m are due to passively sinking particles, with the remaining 30% corresponding to DVM. In that case, the flux attenuation derived from $^{234}$Th would be the same as traps (0.5 mmol C m$^{-2}$ d$^{-1}$), which is within uncertainties with the estimate reported in this study.

**POC Flux from $^{210}$Po**

The uncertainties associated with POC flux estimates from $^{210}$Po at each station accounted for analytical and counting uncertainties following Rigaud et al. (2013). The uncertainty of the cruise-average POC flux attenuation reported here (2.2 ± 0.4 mmol C m$^{-2}$ d$^{-1}$) was calculated by propagating the standard error of the mean (SEM) of the cruise-average POC fluxes at 100 m (2.6 ± 0.3 mmol C m$^{-2}$ d$^{-1}$) and 500 m (0.4 ± 0.2 mmol C m$^{-2}$ d$^{-1}$) obtained from three sampled stations.

Several potential sources of uncertainty are associated with $^{210}$Po flux estimates (Roca-Martí and Puigcorbé, 2024). NSS conditions and physical processes could introduce potential biases into the estimates. Still, the reduced spatial and temporal sampling resolution covered for $^{210}$Po does not allow for their quantification (as done for $^{234}$Th). The fact that $^{210}$Po and $^{234}$Th-derived POC fluxes agree within uncertainties gives confidence in the $^{210}$Po-derived estimates presented here and suggests that POC fluxes were similar in magnitude between June and August-September 2018 (the SS response time for $^{210}$Po was approx. 80 days vs. 20 days for $^{234}$Th; Turnewitsch et al., 2008; Buesseler et al., 2020). However, one important aspect that can be quantified is the potential bias associated with the POC/$^{210}$Po ratio choice. While using the POC/$^{210}$Po ratio in the 1-5 µm size class to estimate POC flux would not change flux attenuation (2.4 ± 0.2 mmol C m$^{-2}$ d$^{-1}$), using the >51 µm size class would result in a two-fold increase in flux attenuation (5.3 ± 0.7 mmol C m$^{-2}$ d$^{-1}$).

**POC Flux from Gel Traps**

A description of the uncertainties associated with POC fluxes estimated by imaging and classifying sinking particles collected in polyacrylamide gel trap collectors can be found in Durkin et al. (2021). Counting uncertainties were calculated by the square root of the number of counts in each particle class and size bin and were propagated to calculate the counting uncertainty associated with the total modeled POC flux of each trap sample. Modeled POC fluxes from multiple traps deployed at the same depth horizon in each epoch were averaged together, and the counting uncertainties were propagated to generate epoch-specific estimates of POC flux. The cruise-wide average modeled POC flux from gel traps was calculated by averaging the three epoch averages and calculating the SEM of those three epochs at each depth horizon. Potential sources of uncertainty associated with gel trap estimates include trap undersampling of particles (discussed above) and those related to modeling particle volumes from images and their conversion to carbon units using published relationships (Durkin et al., 2021).





## Temporal DOC Changes

To be certain that there was a resolvable decrease in the DOC concentration between time points, the instrumental error (± 0.7 µM C) of those time points should not overlap; however, the difference between DOC concentrations was not >1.4 µM for sampling depths >110 m. Therefore, the estimated DOC remineralization rates shown here were not resolved with high confidence. DOC removal rate accuracy is essential in deriving an integrated DOC remineralization rate, which should have been as low as ~2.4 mmol C m$^{-2}$ d$^{-1}$ or as high as 8.9 mmol C m$^{-2}$ d$^{-1}$. At the same time, because the DOC concentrations were

consistently lower at all depths at the end of RR1813, around September 7$^{th}$, 2018, when compared to the July 2018 profiles, this suggests that the DOC concentrations had decreased in the upper mesopelagic zone over time near OSP.

## Chemoautotrophic production

Dissolved inorganic carbon (DIC) fixation estimates have widely been estimated using culture-based estimates of ammonia

and nitrite oxidation using a ratio of 1 to 10 (Reinthaler et al., 2010). This approach was most recently expanded upon for a range of representative ammonia- and nitrite-oxidizing taxa (Bayer et al., 2023), where a similar ratio was found for the combined nitrification process. Field-based nitrification estimates were obtained from profiles spanning the mesopelagic zone during the EXPORTS North Pacific cruise based on $^{15}$N uptake, tracer-based methods (e.g., Santoro et al., 2010; nitrification data currently planned for manuscript submission). In addition to the DIC to nitrification ratio, the primary uncertainty using

this approach to estimate DIC fixation involves the nitrogen-to-carbon conversion factor. In this case, a Redfield-based conversion of 6.6 C:N was used; however, the use of this conversion factor requires further verification.

## Net Community Production

Accuracy and uncertainty in oxygen sensors were determined by inter-calibration across EXPORTS shipboard and autonomous

sensors and ultimately tied to bottle Winkler samples (Traylor et al., submitted). Error estimates provided are the standard deviation of 1000 runs of an oxygen mass balance model (Traylor et al., submitted), with uncertainties applied for the following terms: the O$_2$:C ratio (1.40 ± 0.15), the diffusive gas flux (10%), partially collapsing bubbles (30%), completely collapsing bubbles (30%), diapycnal eddy diffusivity (100%), and oxygen saturation (0.5%). Uncertainty in the gas flux was also addressed by randomly choosing between the gas flux parameterization of Nicholson et al. (2016) and Liang et al. (2013) with

a bubble flux adjustment (Emerson et al., 2019) for each model run. These flux estimates primarily diverge at high wind speeds due to differences in the treatment of the bubble flux. Atmospheric exchange is rapid compared to other physical processes, causing any horizontal oxygen gradients to be minimal in the surface ocean (Emerson et al., 1995). Consequently, horizontal advection and diffusion fluxes were assumed to be negligible compared to other processes and omitted, though this may constitute an additional source of uncertainty. Additionally, the flight path of the Seaglider in the experiment was designed to

follow a Lagrangian framework and to minimize horizontal advective fluxes.

## **Carbon Demand in the Mesopelagic Zone**





**Free-living Bacterial Respiration**

**Carbon Conversion Uncertainty**

Cell carbon conversions presented here resulted in nearly half the bacterial biomass as compared to prior studies for OSP (biovolume data also suggest an average of 11 fg C cell$^{-1}$ as derived in Stephens et al. (2020) vs. 20 fg C cell$^{-1}$ assumed in Kirchman et al. (1993), Sherry et al. (1999), Simon et al. (1992)). Similarly, the empirically derived leucine conversion factors for bacterial production (BP; Kirchman, 1992) used in the current study were, on average, 1.4-fold lower than if using the

more commonly applied carbon conversion factor for BP (e.g., 1.5 kg C pmol Leu$^{-1}$; Simon and Azam, 1989).

**Bacterial Growth Efficiency Uncertainty**

Typical bacterial growth efficiencies (BGE) estimated for sub-euphotic zone waters are <12%, where median values for the oceans are around 8% (4-12% range). However, it is important to note that methods for estimating sub-euphotic zone BGEs

have often combined two independent methods, each requiring its carbon conversion assumptions. For instance, bacterial production rates are calculated using ~3-5-hour incubations with $^3$H-labeled Leucine or $^3$H-labeled Thymidine, each with variable carbon conversion factors. Respiration, on the other hand, is typically measured using oxygen removal rates in oxygen flasks (e.g., Sherry et al. 1999 for OSP) but has also been measured over 3-6 hours using the reduction of tetrazolium salts by respiring organism's electron transport system (e.g., Martínez-García et al. 2009). Respiration methods also require

assumptions of the respiratory quotient (RQ; $O_2$:C) to convert to carbon units. In addition to carbon conversion concerns, the comparison of BP to bacterial respiration (BR) to estimate BGEs can also be influenced by unconstrained contributions of processes other than BR, resulting in changes in oxygen (e.g., organisms other than bacteria can slip through 1.2 µm or even 0.8 µm-sized filters, the RQ can be variable depending on the organic substrate, among other contributions). Using an alternate approach to specifically estimate BGEs, we simultaneously track changes to bacterial carbon and organic carbon. This

approach has the advantage that there are fewer assumptions about carbon conversions.

**Depth-based Uncertainty**

BGEs, in theory, decrease with depth in the oceans due to less available organic and micronutrient resources that would otherwise support more efficient growth by bacteria. In the current study, we have only directly estimated the upper bound of

the mesopelagic zone at 95 m, whereas we are interested in the processes between 95 and 500 m. However, the 95 m depth was below the steepest attenuation for BP and bacterial abundances, so we consider the 95 m representative of the 95 to 500 m zone.

**Particle-attached Bacterial Respiration**

Sources of uncertainty involve oxygen measurements made using the Aanderaa optode 4831 Series (± 2 µM or 1.5%), linear regressions, POC measurements, and a 10% error associated with sample splitting. Error from these uncertainties and





measurements were propagated through all calculations. Potential artifacts of the RESPIRE trap method are described in detail in Boyd et al. (2015). Some of these artifacts include arrested sinking of particles and cryptic zooplankton visits into the RESPIRE incubation chambers that could result in increased rate measurements. A decrease in respiration rates could result

from a lack of fluid motion inside the RESPIRE traps, creating a microenvironment within the incubation chamber. A respiratory quotient of 1.1 was also assumed to convert oxygen consumption rates to carbon units.

**Microzooplankton respiration**

Using a combination of two different pre-filtration schemes in dark dissolved organic matter remineralization incubations,

Stephens et al. (2020) provide predictive evidence of microzooplankton grazing rates on bacterioplankton (combination of archaea, heterotrophic bacteria, and cyanobacteria). The primary assumption in this approach is that differences in the growth rates of bacteria observed in <3 μm filtered incubations and the incubations where <3 μm filtered water was diluted by 0.2 μm filtered water are due to microzooplankton grazing loss. This approach is similar in concept to the Landry and Hasset (1982) dilution method used to estimate phytoplankton grazing rates. Another uncertainty in this approach requires an estimate of

ingestion efficiency for microzooplankton (similar to BGE as described above), with a mean of 40%. Similarly to BGE, variations in this ingestion efficiency value (5-80% potential range) can lead to significant differences in predicted microzooplankton respiration rates.

**Long-term Respiration**

Long-term respiration rates in the mesopelagic were estimated from the biogeochemical float (BGC-float) and Seaglider. Oxygen sensors were calibrated against Winkler oxygen measurements conducted during the EXPORTS shipboard sampling. Calibrated vertical sensor profiles were first interpolated to isopycnal surfaces. Each time series along a surface was then corrected for water-mass variability by subtracting the correlation between isopycnal spice and oxygen from the time series. A rate for each surface was then calculated by linear interpolation of spice-corrected oxygen versus time. Isopycnal coordinates

were converted back to depth coordinates by taking the mean depth of each isopycnal surface. Uncertainty was calculated based on the standard error in slope and integrated in the vertical. The $O_2$-based respiration estimates were converted to carbon units using the Redfield ratio ($O_2$:C=1.4).

**Carbon Supply and Demand**

**Zooplankton metabolic contributions**

There are four major sources of uncertainty regarding the zooplankton dataset. Two of the uncertainties stem from using paired daytime and nighttime tows to estimate the relative contributions of migratory and resident communities. It is known that there is a substantial amount of patchiness in zooplankton distributions and that net avoidance of larger zooplankton during the daytime can skew the assessment of migrator biomass. As this study focuses on the 100-500 m depth bins, where light levels are lower, this may play less of a role than it would in the epipelagic. Additionally, the bulk day-night estimate of migrator






biomass and the classification of all circumstances where a size class had a higher biovolume during the night compared to the day as "resident" organisms may obscure patterns of reverse migration (Hays, 2003; Heywood, 1996). This would only truly modify the estimate of particulate carbon flux, which is about 50% higher, on average, for resident versus migratory individuals.


The second two sources of uncertainty are associated with applying allometric equations to predicting metabolic rate. Our zooplankton biomass is divided into five size fractions, and we have followed the standard protocol of using an average-sized individual within each of these bins to do our allometric estimates of physiology. It is well known, however, that the particle size distribution of organisms follows a power function with a greater number of smaller individuals within each size class.

Metabolism also follows a power law function whereby smaller animals have substantially faster physiological rates per unit of biomass. Together, this means that there are likely many smaller animals with faster metabolism within each size bin than is calculated, resulting in an underestimation of physiological rates (Maas et al., 2021).

The final source of uncertainty is the fact that most of the experiments used to generate the equations applied to predict

metabolic rates are skewed in favor of larger, epipelagic, or migratory species ($p$CO$_2$, DOC, POC), or were validated in taxonomically non-relevant groups (the size-based mortality estimate was originally validated using fish stocks). It is known, however, that depth negatively correlates with the metabolic rate of various groups of meso- and bathypelagic organisms, although neither the ubiquity of this trend across phylum nor the physiological or behavioral causes have yet to be conclusively identified (Ikeda, 2014; Ikeda et al., 2006; Seibel and Drazen, 2007). A correction for depth was not included in our metabolic

estimates, and as such, our "resident" metabolism may be an overestimate of the true metabolic demand of this midwater community. To add to this overestimation of midwater community physiology, we documented a substantial population of ontogenetic migrators (multiple species of Neocalanus spp.) that have already started their diapause in the region. It has been demonstrated that copepods' metabolism is substantially lower in their diapausing state (Baumgartner and Tarrant, 2017). Most of these organisms were likely diapausing below the 100-500 m layer considered in this study, reducing the error associated

with this population.

**Size-fractionated POC Inversion**

The uncertainties in the particle fluxes estimated in our study were derived by propagating the uncertainties in the posterior estimates of POC concentrations and particle cycling parameters, with due consideration for error covariances (Bevington and

Robinson 2003). We found that estimates of model errors, or residuals, are significant, which may arise from factors such as particle cycling processes that are not represented or are misrepresented in the model, unsteadiness, and physical transport (see Amaral et al., 2022 for more details). These residuals suggest that the POC supply fluxes presented here may be underestimated and/or that the demand fluxes may be overestimated.



## Data Availability

All data presented in this manuscript can be found in Appendix Table 1. Raw data can be found at SeaWiFS Bio-optical and Storage System (SeaBASS, http://dx.doi.org/10.5067/SeaBASS/EXPORTS/DATA001).

## Author Contributions

All authors contributed to the experimental conceptual design, data generation, and analysis. BMS, MRM, AEM, VJA, SC, ST, and RPN contributed to the initial manuscript draft, and all co-authors contributed to the manuscript revision.

## Competing Interests

No authors have competing interests.

## Acknowledgements and Funding

We would like to acknowledge the program management, logistical support, and data provided by participants and collaborators in the National Aeronautics and Space Administration (NASA) as part of the EXport Processes in the Ocean from RemoTe Sensing (EXPORTS) program. The EXPORTS program was supported in large part by significant contributions from Ivona Cetinic and Inia Soto Ramos. Co-authors acknowledge NASA EXPORTS awards 80NSSC17K0555 (KB and CBN), 80NSSC17K0662 (MLE), 80NSSC18K0437 (CAC), 80NSSC17K0552 (NC), 80NSSC18K1431 (AES and PB), 80NSSC17K0663 (DPN and ST), 80NSSC17K0716 (SMD and Tatiana A Rynearson), and 80NSSC17K0692 (DAS). We would also like to acknowledge the following EXPORTS program participants who provided valuable discussions about the data: Colleen Durkin, Mark Brzezinski, Uta Passow, Ben Van Mooy, Brian Popp, Dennis Hansell, Jason Graff, Alex Niebergall, Heather McNair, Elisa Romanelli, and Connor Shea. BMS acknowledges support in part from the National Science Foundation (OCE-2023545) and Taiwan's National Science and Technology Council (NSTC 113-2611-M-002-009). MRM acknowledges support from the Beatriu de Pinós Postdoctoral Program (2021-BP-00109), "la Caixa" Foundation (ID 100010434, fellowship code LCF/BQ/PI24/12040022), the ICTA-UAB "María de Maeztu" Program for Units of Excellence funded by the Spanish Ministry of Science, Innovation and Universities (CEX2019-000940-M), and ICTA-UAB MERS (2021 SGR-640) of the Generalitat de Catalunya. DPN and ST also acknowledge support from NOAA GOMO Grant NA19OAR4320074. ST and VJA acknowledge support from the NSF Graduate Research Fellowship. YH acknowledges support from the Fundamental Research Funds for the Central Universities (grant number: 20720240105). The BGC Argo float data and associated analysis were made possible through support from NSF award 2032754 (AJF and PJL). PJL acknowledges support from NSF-OCE 1829614, and OM acknowledges support from NSF-OCE 1829790. This manuscript is PMEL contribution #5640.