# Peer review of "An upper mesopelagic zone carbon budget for the subarctic North Pacific"

_EGUsphere, 2024_

## Referee Comment (RC1)

**Review of Stephens et al. An upper mesopelagic zone carbon budget for the subarctic North Pacific**

This article presents a suite of ship-based and autonomous measurements from the EXPORTS program, examining carbon mass balances in the upper mesopelagic zone of the NE Subarctic Pacific. By combining a range of methods / approaches with different assumptions, strengths / weaknesses and integration time-scales, the authors aim to better constraint on the fate of organic carbon in the top 100 – 500 m of the water column at Ocean Station Papa. Based on a careful and thoughtful analysis of many different data sets, the authors report an apparent imbalance in carbon supply / demand (excess demand), which they justify based on temporal scale imbalances of processes and measurements. Specifically, they suggest that a seasonally-active DOC pool, associated with production occurring prior to their measurements, was required to meet the estimated C demands. Another notable result was the apparent importance of active carbon export associated with zooplankton DVM processes, which have previously been often overlooked in models. Overall, I think this is an interesting and well-executed study, and that the authors do a good job of discussing the limitations, caveats and method-specific assumptions of their work. I do, however, have a few suggestions that I believe would improve the presentation and clarity of the paper.

Specific Comments:

Abstract:

I find the following sentence to be somewhat confusing: *This imbalance could be resolved by particle dynamics influencing timescales of organic carbon utilization prior to the field campaign*

Perhaps simplify / rephrase as: 'resolved by the production and export of organic carbon prior to our measurement period'

Line 45: sentence beginning with NCP is missing a verb. Maybe add 'were' before 'measured'?

Figure 1. Legend isn't clear – I presume that 18 and 19 represent 2018 and 2019, respectively, but that could be more explicit. Small thing, but maybe put in a little ship icon for the OOI cruise, for consistency with the other cruises identified. And maybe stack the bars in the figure by sampling type (cruises together, then mooring, then gliders and float).

Line 163/4. How do you calculate DIC with only pH measured? As I'm sure they know, one other carbonate system parameter has to be estimated or measured. On line 256, the authors mention the CANYON-B alrgorithm. Is this what was used here?

I find it a bit strange to see a mixture of methods and results in the same section. I realize that the study uses some published data, obtained from methods that have been described previously, but I would have found it useful to have at least a short (but systematic) run-down of the different methods. Given the author's focus on the different assumptions / integration time-scales across methods, it would be nice, I think, to have this spelled out explicitly up front. Later in the text, there is a nice presentation of limitations / caveats (Table 1), and I think it should be referenced here. I still had some questions about the methods (see below), that could have been addressed with a bit more detail on methods.

Figure 2. Maybe I'm missing something obvious, but I don't understand how the POC flux can be 10-times higher than NPP.

Line 260. In the $O_2$- based NCP calculation, were their corrections made for non-biological effects on $O_2$ saturation state (e.g. S and T changes and bubble injection?)?

Line 295. How was bacterial growth efficiency measured? On line 585, there is a reference to BGE measurements made by Stephens 2020. Is this what is being referred to here?

Figure 4. This is a nice summary, overall, but I don't understand how NCP can be less than NPP. What am I missing?

Line 523 at the end 'base don' should be 'based on'.

END OF REVIEW

---

## Referee Comment (RC2)

**Review of EGU 2251, An upper mesopelagic zone carbon budget for the subarctic Noth Pacific**

In this study the authors used multiple approaches to constrain the magnitude of the processes contributing to the mesopelagic zone carbon budget as well as its uncertainties. A snapshot of the budget in August, based on ship measurements showed an imbalance between the estimated organic carbon supply and demand. Instead, the amount of organic carbon necessary to cover the estimated carbon demand in late summer must have come from previous production in spring. This finding challenges the idea of a "steady state carbon budget" in the region of study for timescales of weeks. Independent measurements of the same process (e.g., particulate organic carbon flux) as well as measurements over different timescales (e.g., week-long ship-based measurements vs long-term moorings) provide robustness to the estimated budget and its uncertainties.

This is a relevant and robust study worth publishing, as addressing the range of methodological uncertainties to constrain the carbon budget in the mesopelagic region is important in view of rapidly developing technologies that aim to sequester carbon from the atmosphere via enhancement of the biological carbon pump.

**Specific comments:**

L45: The line about NCP seems to be missing a verb: "Net community production (NCP) rates measured during the preceding spring and early summer of 2018 based on long-term mooring estimates of dissolved inorganic carbon concentrations."

L98-100: Since the conditions were typical for late summer, I find this sentence a bit confusing, were temperatures warmer than expected in 2018? or warmer as compared to when?

L152: I would suggest to change "collected" to "selected from" or something similar, to leave no place for ambiguity as to what was newly sampled in EXPORTS and what was already publicly available. Same goes for L156.

L204: although the Po-based POC flux at 500 m was, within uncertainty, similar to the Th method $(1.2 \pm 1.2)$, as per the uncertainty ranges shown in Fig. 2b and 2c.

L241: Is it possible that 53.7 is 56.7 mmolCm3 maybe? (as per Figure S1a) Otherwise I do not understand how the july-sept difference is more than twice the instrumental uncertainty of 1.4mmolCm3.

L279: The authors mention that "the oxygen-based NCP measurements are of higher spatial and temporal coverage" The temporal coverage of the glider seems to be smaller than the mooring, I wonder if the authors mean, resolution, rather than coverage?

L450: typo in the first word "th1e"

Figure 1. Initially, I thought the numbers in the X-axis represented the day of the month

Figure 3. A vertical line for March could potentially help the reader, as this is a time that was considered to demonstrate NCP contributions (L281).

Throughout the text, I had some questions relative to the regional variability of the samples collected by the different platforms. I feel that these could be clarified with a simple map of the location of the different campaigns/platforms (e.g. to add in the supp). These are examples of the questions:

- Were the DOC profiles sampled at the same location? I understand that the three DOC profiles were collected during the OOI cruise, was the OOI cruise also within the eddy that the EXPORT cruise was sampling? Were all the stations less than 70 km apart? (as that is the maximum distance for water movement according to your L451) Is this relevant for your uncertainty assumption of minimal lateral advection?

- I  understood that the 2 cruises were moving because they were following a float. How does this look in space? Do I interpret right that all the epochs represent the same water parcel (because the cruise was following the float)?

---

## Author Comment (AC1)

We would like to thank the reviewer for their thoughtful comments and suggestions. NOTE: In the following, we have used blue text to highlight our responses to the Reviewers' comments.

**REVIEWER 1**

Review of Stephens et al. An upper mesopelagic zone carbon budget for the subarctic North Pacific

This article presents a suite of ship-based and autonomous measurements from the EXPORTS program, examining carbon mass balances in the upper mesopelagic zone of the NE Subarctic Pacific. By combining a range of methods / approaches with different assumptions, strengths / weaknesses and integration time-scales, the authors aim to better constraint on the fate of organic carbon in the top 100 – 500 m of the water column at Ocean Station Papa. Based on a careful and thoughtful analysis of many different data sets, the authors report an apparent imbalance in carbon supply / demand (excess demand), which they justify based on temporal scale imbalances of processes and measurements. Specifically, they suggest that a seasonally-active DOC pool, associated with production occurring prior to their measurements, was required to meet the estimated C demands. Another notable result was the apparent importance of active carbon export associated with zooplankton DVM processes, which have previously been often overlooked in models. Overall, I think this is an interesting and well-executed study, and that the authors do a good job of discussing the limitations, caveats and method-specific assumptions of their work. I do, however, have a few suggestions that I believe would improve the presentation and clarity of the paper.

Specific Comments:

Abstract:
I find the following sentence to be somewhat confusing: This imbalance could be resolved by particle dynamics influencing timescales of organic carbon utilization prior to the field campaign Perhaps simplify / rephrase as: 'resolved by the production and export of organic carbon prior to our measurement period'

Response: We will revise this sentence based on the reviewer's recommendation.

Line 45: sentence beginning with NCP is missing a verb. Maybe add 'were' before 'measured'?

Response: This will be revised to "*Net community production (NCP) rates measured during the preceding spring and early summer of 2018 based on long-term mooring estimates of dissolved inorganic carbon concentrations were higher than those measured during the EXPORTS field campaign.*"

Figure 1. Legend isn't clear – I presume that 18 and 19 represent 2018 and 2019, respectively, but that could be more explicit. Small thing, but maybe put in a little ship icon for the OOI cruise, for consistency with the other cruises identified. And maybe stack the bars in the figure by sampling type (cruises together, then mooring, then gliders and float).

Response: These are good suggestions and will be adapted into a revised Figure 1.

Line 163/4. How do you calculate DIC with only pH measured? As I'm sure they know, one other carbonate system parameter has to be estimated or measured. On line 256, the authors mention the CANYON-B alrgorithm. Is this what was used here?

Response: The reviewer's concern here is understood. We will remove the unnecessary phrase, "allowing for the estimation of dissolved inorganic carbon (DIC) concentrations."

I find it a bit strange to see a mixture of methods and results in the same section. I realize that the study uses some published data, obtained from methods that have been described previously, but I would have found it useful to have at least a short (but systematic) run-down of the different methods. Given the author's focus on the different assumptions / integration time-scales across methods, it would be nice, I think, to have this spelled out explicitly up front. Later in the text, there is a nice presentation of limitations / caveats (Table 1), and I think it should be referenced here. I still had some questions about the methods (see below), that could have been addressed with a bit more detail on methods.

Response: We thank the reviewer for this thoughtful comment, it was an issue we discussed at length prior to manuscript submission. We agree that this could also be better addressed up front, prior to presenting the data, for the reader's reference. Starting on Line 165 of the originally submitted manuscript we have not mentioned methods limitations / caveats. Therefore, we have adapted the reviewer's suggestion to more explicitly spell out in this paragraph that we present limitations / caveats later in the manuscript. We will amend the paragraph starting on Line 165 with the italicized text:

"Several of the datasets presented here have been previously published, and the uncertainties surrounding conversion factors are discussed in those publications (e.g., as highlighted above). Therefore, only a brief description of methods is included here. *However, a detailed comparison of the methods assumptions / limitations can be found in Table 1, along with a brief discussion of how the methods may influence our interpretation of the results.* The present study's goal is to evaluate the combined rates of organic carbon supply and demand and elucidate the implications of carbon conversion factors and uncertainty estimates."

Figure 2. Maybe I'm missing something obvious, but I don't understand how the POC flux can be 10-times higher than NPP.

Response: Thank you for checking about this. The subplots in Figure 2 are only for the sub-euphotic zone (i.e., deeper than 95 m), where NPP rates are inherently low. Depth-integrated 14C-based NPP rates from 0 to 100 m averaged $13.8 \pm 1.9$ mmol C m$^{-2}$ d$^{-1}$, compared with mean $^{234}$Th-based POC flux rates at 100 m of $2.0 \pm 0.6$ mmol C m$^{-2}$ d$^{-1}$. We will add the average integrated euphotic zone NPP value in the introduction at Line 111 for quick reference for the reader as follows: "*Below the EZ, POC sinking fluxes were also relatively low, with an export efficiency of 10-14% (POC flux of 1.4-2.0 mmol C m$^{-2}$ d$^{-1}$ vs. integrated NPP of 13.8 mmol C m$^{-2}$ d$^{-1}$ over 0-100 m), similar to previous late summer estimates at the study site (Buesseler et al., 2020; Estapa et al., 2021).*"

Line 260. In the O2- based NCP calculation, were their corrections made for non-biological effects on O2 saturation state (e.g. S and T changes and bubble injection?)?

Response: Yes, mixed layer NCP estimates were corrected for non-biological effects. The following sentence will be added at Line 260 to ensure readers are made aware of this: *"Corrections for non-biological effects were applied based on changes in temperature and salinity and due to the effects of bubble injection (e.g., Emerson et al., 2019)."*

Line 295. How was bacterial growth efficiency measured? On line 585, there is a reference to BGE measurements made by Stephens 2020. Is this what is being referred to here?

Response: Yes, that is correct. The following sentence will be added at Line 296: *"BGE was estimated based on concurrent increases in bacterial cell carbon and decreases in total organic carbon over time in dark incubations conducted throughout the EXPORTS field campaign."*

Figure 4. This is a nice summary, overall, but I don't understand how NCP can be less than NPP. What am I missing?

Response: NCP better incorporates the effects of heterotrophic respiration and so is a better "export potential" metric to compare with our combination of measured supply terms.

Line 523 at the end 'base don' should be 'based on'.

Response: Thank you for catching this error, it will be revised to "*based on*."

END OF REVIEW

---

## Author Comment (AC2)

We would like to thank the reviewer for their thoughtful comments and suggestions. NOTE: In the following, we have used blue text to highlight our responses to the Reviewers' comments.

**REVIEWER 2**

Review of EGU 2251, An upper mesopelagic zone carbon budget for the subarctic North Pacific

In this study the authors used multiple approaches to constrain the magnitude of the processes contributing to the mesopelagic zone carbon budget as well as its uncertainties. A snapshot of the budget in August, based on ship measurements showed an imbalance between the estimated organic carbon supply and demand. Instead, the amount of organic carbon necessary to cover the estimated carbon demand in late summer must have come from previous production in spring. This finding challenges the idea of a "steady state carbon budget" in the region of study for timescales of weeks. Independent measurements of the same process (e.g., particulate organic carbon flux) as well as measurements over different timescales (e.g., week-long ship-based measurements vs long-term moorings) provide robustness to the estimated budget and its uncertainties.

This is a relevant and robust study worth publishing, as addressing the range of methodological uncertainties to constrain the carbon budget in the mesopelagic region is important in view of rapidly developing technologies that aim to sequester carbon from the atmosphere via enhancement of the biological carbon pump.

Specific comments:

L45: The line about NCP seems to be missing a verb: "Net community production (NCP) rates measured during the preceding spring and early summer of 2018 based on long-term mooring estimates of dissolved inorganic carbon concentrations."

Response: Thank you for catching this mistake. The sentence will be rewritten as follows: "*Net community production (NCP) rates measured during the preceding spring and early summer of 2018 based on long-term mooring estimates of dissolved inorganic carbon concentrations were higher than those measured during the EXPORTS field campaign*"

L98-100: Since the conditions were typical for late summer, I find this sentence a bit confusing, were temperatures warmer than expected in 2018? or warmer as compared to when?

Response: The sentence will be clarified as follows: "*The oceanographic setting encountered during the EXPORTS North Pacific field campaign was typical of late-summer conditions at Ocean Station Papa but captured slightly warmer mixed layer temperatures and lower nitrate concentrations compared to historical data from this site (Siegel et al., 2021).*"

L152: I would suggest to change "collected" to "selected from" or something similar, to leave no place for ambiguity as to what was newly sampled in EXPORTS and what was already publicly available. Same goes for L156.

Response: We will use "obtained" instead of "collected" to avoid confusion.

L152: "*In addition, three profiles for DOC concentrations were obtained on July 27th near OSP from the Ocean Observatories Initiative (OOI)-supported cruise (SR1811).*"

L156: "*Continuous surface observational data from January through December 2018 were obtained from the National Oceanic and Atmospheric Administration Pacific Marine Environmental Laboratory's OOI OSP surface mooring*"

L204: although the Po-based POC flux at 500 m was, within uncertainty, similar to the Th method (1.2 ± 1.2), as per the uncertainty ranges shown in Fig. 2b and 2c.

Response: Agreed. We will rephrase the sentence as follows: "*210Po measurements resulted in cruise-mean POC fluxes that, within uncertainties, were similar to 234Th at 100 m (2.6 ± 0.3 mmol C m-2 d-1, Fig. 2c) and at 500 m (0.4 ± 0.2 mmol C m-2 d-1), leading to a flux attenuation of 2.2 ± 0.4 mmol C m-2 d-1*".

L241: Is it possible that 53.7 is 56.7 mmolCm3 maybe? (as per Figure S1a) Otherwise I do not understand how the july-sept difference is more than twice the instrumental uncertainty of 1.4mmolCm3.

Response: Apologies for the confusion there. Instrumental uncertainty was actually 0.7 mmol C m-3, so twice uncertainty was 1.4 mmol C m-3, which is less than the difference between the July and September values (1.7 mmol C m-3). This will be revised to "*DOC concentration differences for samples collected at 95 m between July and September (53.7 and 52.0 mmol C m-3, respectively) were more than twice the mean instrumental uncertainty (e.g., cruise mean coefficient of variation (CV) was 1.3%, resulting in 2x instrumental uncertainty of 1.4 mmol C m-3 at 95 m).*"

L279: The authors mention that "the oxygen-based NCP measurements are of higher spatial and temporal coverage" The temporal coverage of the glider seems to be smaller than the mooring, I wonder if the authors mean, resolution, rather than coverage?

Response: That is correct. We will replace "coverage" with "resolution": "*Given that oxygen-based NCP measurements are of higher spatial and finer temporal resolution, we only use glider-based integrated (0-100 m) NCP during the cruise period*".

L450: typo in the first word "th1e"

Response: Thanks for catching the typo. We will correct it.

Figure 1. Initially, I thought the numbers in the X-axis represented the day of the month

Response: We will replace 18 with 2018 and 19 with 2019 in the x-axis of Fig. 1 to avoid confusion.

Figure 3. A vertical line for March could potentially help the reader, as this is a time that was considered to demonstrate NCP contributions (L281).

Response: Thanks for the suggestion. We will add a vertical line for March.

Throughout the text, I had some questions relative to the regional variability of the samples collected by the different platforms. I feel that these could be clarified with a simple map of the location of the different campaigns/platforms (e.g. to add in the supp). These are examples of the questions:

Response: Thanks for the suggestion. Maps of the different campaigns and platforms considered in this manuscript were provided in the EXPORTS North Pacific overview paper by Siegel et al. (2021). We have asked permission from Elementa to use Figures 5 and 9 from Siegel et al. (2021) in the Supplemental Material for the reader's reference.

- Were the DOC profiles sampled at the same location? I understand that the three DOC profiles were collected during the OOI cruise, was the OOI cruise also within the eddy that the EXPORT cruise was sampling? Were all the stations less than 70 km apart? (as that is the maximum distance for water movement according to your L451) Is this relevant for your uncertainty assumption of minimal lateral advection?

Response: That is a good question. The collection of OOI cruise DOC profiles were planned by Craig Carlson's project group (UCSB) as part of the EXPORTS project. The samples were collected at Ocean Station Papa, which was the starting location of the EXPORTS field campaign in August. These OOI DOC samples were analyzed in Craig Carlson's lab. The eddy that we sampled during the EXPORTS field campaign was not likely at Ocean Station Papa during the OOI Cruise in July. However, the focus of our comparison in the current study is below the euphotic zone, where the water mass velocities at 100-500 m are much slower and likely to be isolated from any of the surface eddy water movements.

- I understood that the 2 cruises were moving because they were following a float. How does this look in space? Do I interpret right that all the epochs represent the same water parcel (because the cruise was following the float)?

Response: The process ship followed a drifting Lagrangian float deployed at approximately 100 m while the survey ship conducted spatial surveys around the process ship (L150).

We will provide further details on the EXPORTS sampling strategy as follows: "*Operations were conducted in three consecutive time intervals or "Epochs" (Siegel et al., 2021). Epoch 1 spanned August 14th to August 23rd, Epoch 2 spanned August 23rd to August 31st, and Epoch 3 spanned August 31st to September 9th, 2018. Each Epoch began with a positioning of the process ship near the Lagrangian float. The spatial scales covered by the process and survey ships are illustrated in Figures 9 and 10 from Siegel et al. (2021).*"